# Performance Bounds for Policy-Based Average Reward Reinforcement Learning Algorithms

**Yashaswini Murthy**
Electrical and Computer Engineering
University of Illinois Urbana-Champaign
Urbana, IL 61801
ymurthy2@illinois.edu

**Mehrdad Moharrami**
Electrical and Computer Engineering
University of Illinois Urbana-Champaign
Urbana, IL 61801
moharami@illinois.edu

**R. Srikant**
Electrical and Computer Engineering
University of Illinois Urbana-Champaign
Urbana, IL 61801
rsrikant@illinois.edu

## Abstract

Many policy-based reinforcement learning (RL) algorithms can be viewed as instantiations of approximate policy iteration (PI), i.e., where policy improvement and policy evaluation are both performed approximately. In applications where the average reward objective is the meaningful performance metric, discounted reward formulations are often used with the discount factor being close to 1, which is equivalent to making the expected horizon very large. However, the corresponding theoretical bounds for error performance scale with the square of the horizon. Thus, even after dividing the total reward by the length of the horizon, the corresponding performance bounds for average reward problems go to infinity. Therefore, an open problem has been to obtain meaningful performance bounds for approximate PI and RL algorithms for the average-reward setting. In this paper, we solve this open problem by obtaining the first finite-time error bounds for average-reward MDPs, and show that the asymptotic error goes to zero in the limit as policy evaluation and policy improvement errors go to zero.

## 1   Introduction

Reinforcement Learning algorithms can be broadly classified into value-based methods and policy-based methods. In the case of discounted-reward Markov Decision Processes (MDPs), value-based methods such as Q-learning [WD92, Tsi94, JJS93, SB18, BT96], fitted value iteration [MS08] and target network Q-learning [CCM22] can be viewed as approximately solving the fixed point of the Bellman optimality equation to find the value function and an approximately optimal control policy. In other words value iteration is approximately implemented for discounted reward MDPs [Ber11, Ber12, Put14]. In policy-gradient methods [AKLM21], a gradient step is used to improve the policy, somewhat similar to the policy improvement step in policy iteration for discounted-reward MDPs. In a recent paper [CM22], it has been shown that many policy-based methods for discounted-reward MDPs can be viewed as special cases of approximate policy iteration. The classical results on approximate policy iteration [BT96] assume that the error in the policy evaluation and improvement steps are constant, independent of the iteration. The key idea in [CM22] is to show that a simple modification of the proof in [BT96] can be used to allow for iteration-dependent policy evaluation and improvement errors, which is then used to make the connection to policy-based methods. Our

37th Conference on Neural Information Processing Systems (NeurIPS 2023).

goal in this paper is to derive similar results for average-reward problems. Average-reward problems [ABB01, WNS21, ZWSW21, Mah96, YB09, Gos04, Sin94] are, in some sense, harder to study than their discounted-reward counterparts. We now discuss why this is so by recalling the error bounds for discounted-reward problems and examining them in an appropriate limit to study their applicability to average-reward problems.

The fundamental result on approximate policy iteration for discounted reward MDPs which allows the connection to policy-based methods is the following [BT96]:

$$\limsup_{k \to \infty} \|J_{\mu_k} - J_*^\alpha\|_\infty \le H_\alpha^2(\epsilon + 2\alpha\delta), \tag{1}$$

where $\alpha$ is the discount factor, $H_\alpha = 1/(1-\alpha)$, $J_*^\alpha$ is the optimal value function, $J_{\mu_k}$ is the value function associated with the policy obtained after $k$ iterations of approximate policy iteration, $\epsilon$ is the policy improvement error, and $\delta$ is the policy evaluation error. Thus, as $\epsilon, \delta \to 0$, we recover the result that standard policy iteration converges. On the other hand, to understand whether the above bound is useful to study average-reward problems, we write Equation (1) as

$$\limsup_{k \to \infty} \frac{1}{H_\alpha} \|J_{\mu_k} - J_*^\alpha\|_\infty \le H_\alpha(\epsilon + 2\alpha\delta).$$

Under mild conditions [Ber11, Ros14], it is well known that each element of the value function vector approaches the average-reward for each fixed policy and for the optimal policy, in the limit $\alpha \to 1$. Hence, the left-hand side of the above equation is an approximation to the error in approximate policy iteration for average-reward MDPs; the right-hand side which gives an upper bound on this error blows up to infinity, i.e., when $\alpha \to 1$. Thus, unlike in the discounted-reward case, the above bound fails to even recover the well-known convergence of standard policy iteration in average-reward case in the limit $\epsilon, \delta \to 0$ (since we have to let $\alpha \to 1$ before letting $\epsilon, \delta \to 0$ ). However, as mentioned in [BT96], it is also well known that approximate policy iteration performs much better than the bound suggests. The main goal of our paper is to resolve this discrepancy between theory and practice.

## 1.1 Contributions

It is well known that the error bound for discounted-reward, approximate policy iteration is tight for unichain MDPs [BT96]. Thus, it is impossible to improve upon the bound in general. However, as will be explained in a later section, it is easy to convert most reasonable unichain MDPs to MDPs where every stationary policy results in an irreducible Markov chain with an arbitrarily small loss of reward. So the natural question to ask is whether the bound can be dramatically improved for MDPs where every stationary policy results in an irreducible Markov chain. To the best of our knowledge, no such bound was available in the prior literature.

Our main contributions are as follows:

- Under the assumption that every stationary policy induces an irreducible Markov chain, we first perform a Schweitzer transformation of the MDP [Sch71] and obtain finite-time error bounds for average-reward approximate policy iteration.

- Using the above finite-time error bounds, we prove an error bound of the form

$$\limsup_{k \to \infty} J^* - J_{\mu_k} \le f(\epsilon, \delta),$$

  for average-reward approximate policy iteration for a function $f$ such that $f(\epsilon, \delta) \to 0$ in the limit as $\epsilon, \delta \to 0$. Note that this is in sharp contrast to the result that one can obtain from the discounted-reward analysis where the error bound blows up to infinity when applied to discount factors approaching $1$, which is the appropriate limit to convert discounted-reward problems to average-reward problems.

- The main difficulty in obtaining the above bound when compared to the discounted-reward case is the lack of infinity-norm contraction property for the Bellman operator in the average-reward case. In most analysis of average-reward problems, this difficulty is circumvented by the use of a span-contraction property [Put14]. However, the span contraction property is insufficient for our purpose and therefore, we use a technique due to [VdW80] for the study of a different algorithm called modified policy iteration in [Put14].

- Next, we extend the above analysis to obtain finite-iteration error bounds for the case where the policy evaluation and improvement errors can potentially vary from iteration to iteration. Further, we allow these errors to be random (as would be the case, for example, when one uses TD learning for policy evaluation and soft policy improvement) and obtain expected error bounds on approximate policy iteration after a finite number of iterations.

- Our error bounds are in a form such that one can then apply them to many policy-based RL algorithms, i.e., we can simply plug in error bounds for specific policy evaluation or policy improvement algorithms. We illustrate the applicability of the results to several different policy-based RL methods (softmax update, greedy update and mirror descent update) studied in the literature [MXSS20, BT96, AYBB$^+$19, AKLM21].

- While the main focus of our paper is on offline learning, we provide connections to regret bounds for online learning in average-reward MDPS as in [AYBB$^+$19, LYAYS21].

## 1.2   Related Work

Our work in this paper is most closely related to the work in [CM22]. The main contribution there was to recognize that an extension of the approximate policy iteration results in [BT96] can be used to derive finite-time error bounds for many RL-based algorithms. The idea of considering iteration-dependent policy evaluation and improvement bounds was also used to study policy iteration with function approximation in discounted-reward MDPs in [WLLS21]. Our contribution here is to derive the first error bound for approximate policy iteration for average-reward MDPs and then write it in a form which can be applied to policy-based RL methods for average-reward TD learning. To the best of our knowledge, no known error bounds existed in prior literature for average-reward approximate policy iteration due to the fact that the corresponding bounds for discounted MDPs increase proportional to the square of length of the horizon. Thus, in the limit as the horizon increases to infinity, we have essentially reduced the dependence of the error from the square of the length of the horizon to linear in the length of the horizon.

RL algorithms have not been studied as extensively for the average-reward case as they are for discounted-reward MDPs. There is recent work [ZZM21] on average-reward TD-learning which we leverage to illustrate the applicability of our results. There are also asymptotic convergence results for policy gradient and actor-critic methods for average-reward MDPs [TVR99, KT99, BSGL09, MMT00]. However, these results only prove convergence to a stationary point and there are no global convergence results. The original paper on natural policy gradient (NPG) is written for average-reward MDPs, but the extensive performance analysis of NPG in subsequent work such as [AKLM21] seems to only consider the discounted-reward case. Recent work [AYBB$^+$19, LYAYS21] considers mirror descent in average reward RL but do not provide performance results for other RL algorithms. In a later section, we compare our results to [AYBB$^+$19, LYAYS21] in the special case of mirror descent-type policy updates

## 2   Model and Preliminaries

### 2.1   Average Reward Formulation

We consider the class of infinite horizon MDPs with finite state space $\mathcal{S}$, finite action space $\mathcal{A}$, and transition kernel $\mathbb{P}$, where $|\mathcal{S}| = n$ and $|\mathcal{A}| = m$. Let $\Delta\mathcal{A}$ denote the probability simplex over actions. We consider the class of randomized policies $\Pi = \{\mu : \mathcal{S} \to \Delta\mathcal{A}\}$, so that a policy $\mu$ assigns a probability vector over actions to each state. Given a policy $\mu$, the transition kernel for the underlying Markov process is denoted by $\mathbb{P}_\mu : \mathcal{S} \to \mathcal{S}$, where $\mathbb{P}_\mu(s'|s) := \sum_{a \in \mathcal{A}} \mu(a|s)\mathbb{P}(s'|s, a)$ is the probability of moving to state $s' \in \mathcal{S}$ from $s \in \mathcal{S}$ upon taking action $\mu(s) \in \mathcal{A}$. Associated with each state-action pair $(s, a)$, is a one-step reward which is denoted by $r_\mu(s) := \mathbb{E}_{a \sim \mu(s)} r(s, a)$.

Let $J_\mu \in \mathbb{R}$ be the average reward associated with the policy $\mu \in \Pi$, i.e. $J_\mu$ is defined as:

$$J_\mu = \lim_{T \to \infty} \frac{\mathbb{E}_\mu \left( \sum_{i=0}^{T-1} r_\mu(s_i) \right)}{T}.$$

Here the expectation is taken with respect to the measure $\mathbb{P}_\mu$ associated with the policy $\mu$. Let $h_\mu \in \mathbb{R}^n$ be the relative value function associated with the policy $\mu$. Defining $\mathbf{1} \in \mathbb{R}^n$ to be the

vector of all ones, the pair $(J_\mu, h_\mu)$ satisfies the following average reward Bellman equation:

$$J_\mu \mathbf{1} + h_\mu = r_\mu + \mathbb{P}_\mu h_\mu. \tag{2}$$

Let $\pi_\mu \in \mathbb{R}^n$ denote the stationary distribution associated with the kernel $\mathbb{P}_\mu$. Let $\mathbb{P}_\mu^* = \mathbf{1}\pi_\mu^\top \in \mathbb{R}^{n \times n}$ denote the matrix whose rows are $\pi_\mu^\top$. Using $\mathbb{P}_\mu^* = \mathbb{P}_\mu^* \mathbb{P}_\mu$, we get the following characterization of the average reward by multiplying both sides of Equation (2) with $\mathbb{P}_\mu^*$: $J_\mu \mathbf{1} = \mathbb{P}_\mu^* r_\mu$.

Let $J^* := \max_{\mu \in \Pi} J_\mu$ be the optimal average reward. From standard MDP theory, there exists $h^* \in \mathbb{R}^n$, for which the pair $(J^*, h^*)$ satisfies the following Bellman optimality equation:

$$J^* \mathbf{1} + h^* = \max_{\mu \in \Pi} r_\mu + \mathbb{P}_\mu h^*. \tag{3}$$

Let $\mu^*$ be the optimizing policy in Equation (3). Then, similar reasoning shows that $J^* \mathbf{1} = \mathbb{P}_{\mu^*}^* r_{\mu^*}$.

The goal of dynamic programming and reinforcement learning is to determine this optimal average reward $J^*$ and its corresponding optimal policy $\mu^*$. Policy iteration is one of the most widely used dynamic programming algorithms for this purpose. It consists of two steps: (i) evaluation of the value function associated with a policy, and (ii) determining a greedy policy with respect to this evaluation. There are primarily two challenges that arise when applying such an algorithm: (i) high memory and time complexity when the state space is too large, and (ii) the unknown transition probability kernel that governs the dynamics of the MDP.

When the state space is too large, it may be computationally infeasible to perform policy iteration exactly. Existing literature suggests that approximate policy iteration in the context of discounted reward MDPs yields good performance. However, such an algorithm has not been studied in the context of average reward MDPs. In fact, the results obtained for the discounted reward MDPs yield vacuous performance bounds for average reward MDPs given the standard relationship between discounted reward and average reward MDPs. We explore this further in Section 2.2.

## 2.2 Relationship to the Discounted Reward MDP

To provide some background, we first discuss the approximate policy iteration algorithm in the context of discounted reward MDPs. Let $\alpha \in [0, 1)$ denote the discount factor. The discounted reward associated with a deterministic policy $\mu \in \Pi$ starting from some state $s \in \mathcal{S}$ is denoted $J_\mu^\alpha(s)$ and is defined as:

$$J_\mu^\alpha(s) = \mathbb{E}_\mu \left[ \sum_{i=0}^\infty \alpha^i r(s_i, \mu(s_i)) \Big| s_0 = s \right].$$

The value function $J_\mu^\alpha \in \mathbb{R}^n$ is the unique fixed point of the Bellman operator $\mathsf{T}_\mu^\alpha : \mathbb{R}^n \to \mathbb{R}^n$ associated with the policy $\mu$, which is defined as $\mathsf{T}_\mu^\alpha J = r_\mu + \alpha \mathbb{P}_\mu J$. For each state $s \in \mathcal{S}$, let $J_*^\alpha(s)$ denote the optimal discounted reward, i.e., $J_*^\alpha(s) = \max_{\mu \in \Pi} J_\mu^\alpha(s)$. Similarly, $J_*^\alpha \in \mathbb{R}^n$ is the unique fixed point of the optimality Bellman operator $\mathsf{T}^\alpha : \mathbb{R}^n \to \mathbb{R}^n$, which is defined as $\mathsf{T}^\alpha J = \max_{\mu \in \Pi} (r_\mu + \alpha \mathbb{P}_\mu J)$. Algorithm 1 is Approximate Policy Iteration for the discounted reward MDP:

---

**Algorithm 1** Approximate Policy Iteration: Discounted Reward

---

Require $J_0 \in \mathbb{R}^n$
**for** $k = 0, 1, 2, \ldots$ **do**
    1. Compute $\mu_{k+1} \in \Pi$ such that $\|\mathsf{T}^\alpha J_k - \mathsf{T}_{\mu_{k+1}}^\alpha J_k\|_\infty \leq \epsilon$   ▷ Approximate Policy Improvement
    2. Choose $J_{k+1}$ such that $\|J_{k+1} - J_{\mu_{k+1}}\|_\infty \leq \delta$       ▷ Approximate Policy Evaluation
       where $J_{\mu_{k+1}} = \mathsf{T}_{\mu_{k+1}}^\alpha J_{\mu_{k+1}}$
**end for**

---

**Theorem 2.1.** *Let $\mu_k$ be the sequence of policies generated from the approximate policy iteration algorithm (Algorithm 1). Then the performance error is bounded as:*

$$\limsup_{k \to \infty} \|J_{\mu_k} - J_*^\alpha\|_\infty \leq \frac{\epsilon + 2\alpha\delta}{(1 - \alpha)^2}. \tag{4}$$

*Proof.* The proof of this theorem can be found in [Ber11]             □

From literature [Ros14, Ber11], we know that the average reward $J_\mu$ associated with any policy $\mu$ is related to its discounted reward $J_\mu^\alpha(s)$ counterpart as follows:

$$J_\mu = \lim_{\alpha \to 1} (1 - \alpha) J_\mu^\alpha(s).$$

Note that the above relation is independent of the state $s \in \mathcal{S}$, as the average reward does not depend on the initial state. Multiplying Equation (4) with $(1 - \alpha)$, and letting $\alpha \to 1$, still yields an approximation performance bound with $(1 - \alpha)$ in the denominator. This term blows up to infinity as $\alpha \to 1$. Note that this bound is known to be tight under unichain Markov structure of the probability transition kernel [Ber11]. However, in practice it is observed that approximate policy iteration works well in the average reward MDP case although these theoretical bounds are not representative of this performance. To the best of our knowledge, we are unaware of an average reward approximate policy iteration performance bound, when the policies induce an irreducible Markov chain. We bridge this gap between theory and practice by providing a theoretical analysis of approximate policy iteration in the average reward MDP setting, with non trivial performance bounds.

## 3 Approximate Policy Iteration for Average Reward

A crucial component of the proof of convergence of approximate policy iteration in the context of discounted reward MDPs is the contraction due to the discount factor $\alpha$ which is absent in the average reward setting (since $\alpha = 1$). To get some source of contraction, we have to make some assumptions on the MDP.

**Assumption 3.1.** We make the following assumptions:

(a) Every deterministic policy $\mu \in \Pi$ induces an irreducible Markov Chain $\mathbb{P}_\mu$.

(b) For all policies, the diagonal elements of the probability transition matrix are positive, i.e., the Markov chain stays in the same state with non-zero probability.

Assumption (a) need not be satisfied by all MDPs. However, in order to satisfy this assumption, we consider a modified MDP where at every time step with probability $\varepsilon$, an action is chosen from the set of all possible actions with equal probability. Simultaneously, with probability $1 - \varepsilon$, we choose an action dictated by some policy. The problem then is to choose this policy optimally. For most MDPs of interest, this small modification will satisfy our assumption with a small $O(\varepsilon)$ loss in performance. It is straightforward to show the $O(\varepsilon)$ loss, but we include the proof in the appendix for completeness. Assumption (b) is without loss of generality in the following sense: there exists a simple transformation which essentially leaves the MDP unchanged but ensures that this assumption is satisfied. This transformation known as the aperiodicity transformation or Schweitzer transformation was introduced in [Sch71]. For more details, the reader is referred to the appendix. In the remainder of this paper, we assume that all MDPs are transformed accordingly to ensure the assumptions are satisfied. One consequence of Assumption (b) is the following lemma which will be useful to us later.

**Lemma 3.2.** *There exists a $\gamma > 0$ such that, for any policy $\mu \in \Pi$,*

$$\min_{i \in \mathcal{S}} \pi_\mu(i) \geq \gamma,$$

*where $\pi_\mu$ is stationary distribution over states associated with the policy $\mu$.*

*Proof.* This lemma has been proved in a more general sense in the context of deterministic policies in [VdW80]. Since the aperiodicity transformation ensures a non-zero probability of staying in the same state, the expected return times are bounded and the lemma holds true for randomized policies as well. □

Prior to presenting the algorithm, consider the following definitions. The average-reward Bellman operator $\mathsf{T}_\mu : \mathbb{R}^n \to \mathbb{R}^n$ corresponding to a policy $\mu$ is defined as $\mathsf{T}_\mu h = r_\mu + \mathbb{P}_\mu h$. The average-reward optimal Bellman operator $\mathsf{T} : \mathbb{R}^n \to \mathbb{R}^n$ is defined as $\mathsf{T} h = \max_{\mu \in \Pi} r_\mu + \mathbb{P}_\mu h$. The value function $h_\mu$ associated with a policy $\mu$ satisfies the Bellman Equation (2) with single step reward $r_\mu$ and the transition kernel $\mathbb{P}_\mu$. Note that $h_\mu$ is unique up to an additive constant.

Solving for $(J_\mu, h_\mu)$ using Equation (2) involves setting the value function for some fixed state $x^*$ as 0 (since this reduces the system from being underdetermined to one with a unique solution). Further, the value function $h_\mu$ can be alternatively expressed as the $\lim_{m\to\infty} \mathsf{T}_\mu^m h_0$ for any $h_0 \in \mathbb{R}^n$ as is the case with discounted reward MDPs. However, unlike the discounted reward Bellman Equation, there is no discount factor $\alpha < 1$ preventing the value function from exploding to infinity. Hence, we consider value function computed using the relative Bellman operator $\widetilde{\mathsf{T}}_\mu$ defined as

$$\widetilde{\mathsf{T}}_\mu h = r_\mu + \mathbb{P}_\mu h - r_\mu(x^*)\mathbf{1} - (\mathbb{P}_\mu h)(x^*)\mathbf{1}.$$

We now state the algorithm for Approximate Policy Iteration for Average Reward MDPs.

### 3.1 Approximate Policy Iteration Algorithm for Average Reward MDPs

---
**Algorithm 2** Approximate Policy Iteration: Average Reward
---
1: Require $h_0 \in \mathbb{R}^n$
2: **for** $k = 0, 1, 2, \ldots$ **do**
3:      1. Compute $\mu_{k+1} \in \Pi$ such that $\|\mathsf{T}h_k - \mathsf{T}_{\mu_{k+1}} h_k\|_\infty \le \epsilon$      ▷ Approx. Policy Improvement
4:      2. Compute $h_{k+1}$ such that $\|h_{k+1} - h_{\mu_{k+1}}\|_\infty \le \delta$      ▷ Approx. Policy Evaluation
5:          where $h_{\mu_{k+1}} = \lim_{m\to\infty} \widetilde{\mathsf{T}}_{\mu_{k+1}}^m h_k$
6: **end for**
---

### 3.2 Performance bounds for Average Reward Approximate Policy Iteration

We are now ready to present the main result of this work.

**Theorem 3.3.** *Let $J^*$ be the optimal average reward of an MDP that satisfies Assumption 3.1. The sequence of policies $\mu_k$ generated by Algorithm 2 and their associated average rewards $J_{\mu_k}$ satisfy the following bound:*

$$\left(J^* - J_{\mu_{k+1}}\right) \le \underbrace{\frac{\left(1 - (1-\gamma)^k\right)}{\gamma} \left(\epsilon\left(\gamma + 1\right) + 2\delta\right)}_{\text{approximation error}} + \underbrace{(1-\gamma)^k \left(J^* - \min_i \left(\mathsf{T}h_0 - h_0\right)(i) + \epsilon\right)}_{\text{initial condition error}}.$$

**Interpretation of the Bound:** Since $0 < \gamma < 1$, as $k \to \infty$, the error due to initial condition $h_0$ drops to zero. The approximation error consists of two components, $\epsilon$ and $\delta$. Here, $\epsilon$ represents the approximation error due to a suboptimal policy and $\delta$ represents the approximation error associated with evaluating the value function corresponding to a policy. The limiting behavior of the average reward associated with the policies obtained as a consequence of the algorithm is captured in the following corollary.

**Corollary 3.4.** *The asymptotic performance of the policies obtained from Algorithm 2 satisfies:*

$$\limsup_{k\to\infty} \left(J^* - J_{\mu_k}\right) \le \frac{(1+\gamma)\epsilon + 2\delta}{\gamma}. \tag{5}$$

Note that the asymptotic bound in Equation (5) is in a very similar form as the asymptotic performance bound for the discounted reward MDP in Equation (4) where $\gamma$ plays the role of $(1 - \alpha)^2$. However, when $\epsilon$ and $\delta$ got to zero, this bound also goes to zero which is not the case if we try to approximate average-reward problems by using the results for discounted-reward problems in the limit where the horizon goes to infinity.

## 4 Application to Reinforcement Learning

Approximate policy iteration is closely related to RL. In general, approximate policy improvement and approximate policy evaluation in Algorithm 2 are executed through approximations to policy improvement (such as greedy update, mirror descent, softmax) and TD learning for value function approximation. First, we present a generic framework to analyze policy-based RL algorithms by

building upon the theory presented in the last section. For this purpose, we define the state-action relative value function $Q$ (instead of the relative state value function $h$ as before) to evaluate a policy. Let $\mathsf{T}^{\mathsf{Q}}_\mu Q$ denote the Bellman operator with respect to $Q$ and policy $\mu$ where $(\mathsf{T}^{\mathsf{Q}}_\mu Q)(s,a) = r(s,a) + (\mathbb{Q}_\mu Q)(s,a)$, where $\mathbb{Q}_\mu(s',a'|s,a) = \mu(a'|s')\mathbb{P}(s'|s,a)$. The relative state action value function corresponding to policy $\mu$ is represented by $Q_\mu$ and is the solution to the following Bellman equation: $J_\mu + Q_\mu(s,a) = r(s,a) + \sum_{s'\in\mathcal{S},a'\in\mathcal{A}} \mathbb{Q}_\mu(s',a'|s,a)Q_\mu(s',a')$, for all state action pairs $(s,a) \in (\mathcal{S},\mathcal{A})$. We present a generic policy-based algorithm for average-reward problem below.

---

**Algorithm 3** Generic Policy Based Algorithm: $Q$-function Average Reward

---

    Require $Q_0 \in \mathbb{R}^{|\mathcal{S}||\mathcal{A}|}$
    **for** $k = 0, 1, 2, \ldots, T$ **do**
        1. Determine $\mu_{k+1} \in \Pi$ as a function of $Q_k$ using a possibly random policy improvement step
        2. Compute $Q_{k+1}$ as an approximation to $Q_{\mu_{k+1}}$ using (state, action, reward) samples from a
    trajectory generated by policy $\mu_{k+1}$
    **end for**

---

Note that the error in Steps 1 (policy improvement) and 2 (policy evaluation) of the above algorithm could be random. Thus, the analysis for Theorem 3.3 has to be adapted to Algorithm 3. The resulting expected deviation from optimality is characterized in the lemma below.

**Lemma 4.1.** *Let $\mu_k$ and $Q_k$ be the sequence of policies and relative state-action value function iterates generated by Algorithm 3. For all $k \in 0, \ldots, T$, we have:*

$$\left(J^* - \left(\min_{(s,a)}\left(\mathsf{T}^{\mathsf{Q}}Q_k - Q_k\right)(s,a)\right)\right) \leq (1-\gamma)\left(J^* - \left(\min_{(s,a)}\left(\mathsf{T}^{\mathsf{Q}}Q_{k-1} - Q_{k-1}\right)(s,a)\right)\right)$$
$$+ \left\|\mathsf{T}^{\mathsf{Q}}Q_{k-1} - \mathsf{T}^{\mathsf{Q}}_{\mu_k}Q_{k-1}\right\|_\infty + 2\|Q_k - Q_{\mu_k}\|_\infty,$$

*where*

$$\gamma = \min_{\substack{\mu\in\mu_1,\mu_2,\ldots,\mu_T \\ (s,a)\in\mathcal{S}\times\mathcal{A}}} \mathbb{Q}^*_\mu(s,a) > 0,$$

*and $\mathbb{Q}^*_\mu$ is a matrix with identical rows corresponding to the invariant distribution of $\mathbb{Q}_\mu$.*

Iteratively applying Lemma 4.1 yields the following proposition for a generic policy-based algorithm.

**Proposition 4.2.** *For any $T > 0$, the iterates of Algorithm 3 satisfy*

$$\mathbb{E}\left[J^* - J_{\mu_{T+1}}\right] \leq \underbrace{(1-\gamma)^T \mathbb{E}\left[J^* - \min_i\left(\mathsf{T}^{\mathsf{Q}}Q_0 - Q_0\right)(i)\right]}_{\text{Initial condition error}} + \underbrace{2\sum_{\ell=0}^{T-1}(1-\gamma)^\ell \mathbb{E}\left[\left\|Q_{T-\ell} - Q_{\mu_{T-\ell}}\right\|_\infty\right]}_{\text{Policy evaluation error}}$$

$$+ \underbrace{\sum_{\ell=1}^{T-1}(1-\gamma)^{\ell-1}\mathbb{E}\left[\left\|\mathsf{T}^{\mathsf{Q}}_{\mu_{T+1-\ell}}Q_{T-\ell} - \mathsf{T}^{\mathsf{Q}}Q_{T-\ell}\right\|_\infty\right] + \mathbb{E}\left[\left\|\mathsf{T}^{\mathsf{Q}}_{\mu_{T+1}}Q_T - \mathsf{T}^{\mathsf{Q}}Q_T\right\|_\infty\right]}_{\text{Policy improvement error}}.$$

We now illustrate the applicability of the above result to specific RL algorithms. More specifically, we characterize finite time performance bounds when state-action relative value functions are learnt through TD-Learning [ZZM21], while several different update rules are employed for policy improvement.

## 4.1 Performance bounds for RL algorithms

For every iteration of Algorithm 3, it is necessary to observe trajectories of length $\tau$ in order to learn the state action value function associated with any policy. Using linear function approximations for value functions, the error in policy evaluation at iteration $k$ in Algorithm 3 can be expressed in the following form:

$$\|Q_k - Q_{\mu_k}\|_\infty \leq \|\Phi(\theta_k - \theta^*_{\mu_k})\|_\infty + \|Q_{\mu_k} - \Phi\theta^*_{\mu_k}\|_\infty, \tag{6}$$

where $\Phi \in \mathbb{R}^{|\mathcal{S}||\mathcal{A}| \times d}$ corresponds to the feature vector matrix and $\theta \in \mathbb{R}^d$ is the parameter vector. Further, $\theta_k$ is the parameter vector obtained as a result of the TD learning algorithm. The projection of $Q_{\mu_k}$ onto the span of $\Phi$ is given by $\Phi\theta_{\mu_k}^*$. The last term in Equation (6) corresponds to the function approximation error which depends on the span of the feature matrix $\Phi$. We denote this error term by $\delta_{0,k}$. The error due to TD learning (first term in Equation (6)) is denoted by $\delta_{\text{TD},k}$. Assuming $\max_{(s,a) \in \mathcal{S} \times \mathcal{A}} \|\phi(s,a)\|_2 \leq 1$, and $\|\theta_{\mu_k}^*\|_2$ is bounded uniformly in $k$, [ZZM21] shows that there exists $C > 0$ such that the length of the sample trajectory $\tau$ and $\delta_{\text{TD},k}$ are related as follows:

$$\mathbb{E}\left[\delta_{\text{TD},k}\right] = \frac{C}{\sqrt{\tau}}. \tag{7}$$

Let $c_0 = \left(J^* - \min_{i \in \mathcal{S}} \left(\mathsf{T}^{\mathsf{Q}} Q_0 - Q_0\right)(i)\right)$ be the error due to initial condition and $\overline{\delta}_0 = \max_t \delta_{0,t}$ be the maximum function approximation error across all iterations. Then we obtain the following performance bounds when TD learning is used for policy evaluation.

**Corollary 4.3.** *(Greedy policy update) Let $\tilde{a}(s) = \text{argmax}_{a \in \mathcal{A}} Q_k(s, a')$. Let $\beta > 0$ be such that the sequence of policies in Algorithm 3 corresponding to all state action pairs $(s, a)$ are obtained using the following algorithm, which we call the greedy policy update.*

$$\mu_{k+1}(a|s) = \begin{cases} \frac{1}{\beta|\mathcal{A}|}, & \text{if } a \neq \tilde{a}(s) \\ \frac{1}{\beta|\mathcal{A}|} + 1 - \frac{1}{\beta}, & \text{if } a = \tilde{a}(s), \end{cases} \tag{8}$$

*Let $\eta = \max_{\substack{s \in \mathcal{S}, a \in \mathcal{A} \\ k \in 1 \ldots T}} |Q_k(s, a)|$. Using TD Learning with linear value function approximation for policy evaluation, we get the following finite time performance bound:*

$$\mathbb{E}\left[J^* - J_{\mu_{T+1}}\right] \leq (1 - \gamma)^T c_0 + \left(\frac{1+\gamma}{\gamma}\right)\frac{2\eta}{\beta} + \frac{2}{\gamma}\left(\overline{\delta}_0 + \frac{C}{\sqrt{\tau}}\right). \tag{9}$$

**Corollary 4.4.** *(Softmax policy update) Let $\beta > 0$ be such that the sequences of policies in Algorithm 3 corresponding to all state action pairs $(s, a)$ are obtained using the following algorithm, which we call the softmax policy update.*

$$\mu_{k+1}(a|s) = \frac{\exp\left(\beta Q_k(s, a)\right)}{\sum_{a' \in \mathcal{A}} \exp\left(\beta Q_k(s, a')\right)}. \tag{10}$$

*Using TD Learning with linear value function approximation for policy evaluation, we get the following finite time performance bound:*

$$\mathbb{E}\left[J^* - J_{\mu_{T+1}}\right] \leq (1 - \gamma)^T c_0 + \left(\frac{1+\gamma}{\gamma}\right)\frac{\log\left(|\mathcal{A}|\right)}{\beta} + \frac{2}{\gamma}\left(\overline{\delta}_0 + \frac{C}{\sqrt{\tau}}\right). \tag{11}$$

**Corollary 4.5.** *(Mirror descent update) Let $\beta > 0$ be such that the sequences of policies in Algorithm 3 corresponding to all state action pairs $(s, a)$ are obtained using the following algorithm, which we call the mirror descent policy update.*

$$\mu_{k+1}(a|s) \propto \mu_k(a|s)\exp\{\beta Q_k(s, a)\}. \tag{12}$$

*Let $\omega = \min_{\substack{k \in 1 \ldots T \\ s \in \mathcal{S}}} \mu_k(a^*(s)|s)$ where $a^*(s)$ is the optimal action at state $s$. Using TD Learning with linear value function approximation for policy evaluation, we get the following finite time performance bound:*

$$\mathbb{E}\left[J^* - J_{\mu_{T+1}}\right] \leq (1 - \gamma)^T c_0 + \left(\frac{1+\gamma}{\gamma\beta}\right)\log\left(\frac{1}{\omega}\right) + \frac{2}{\gamma}\left(\overline{\delta}_0 + \frac{C}{\sqrt{\tau}}\right). \tag{13}$$

*Remark* 4.6. The following remarks are in order:

- In each of the above performance bounds, there is an inevitable constant error $\left(\overline{\delta}_0\right)$ due to function approximation. If the value function lies in the class of functions chosen, then this error will be zero.

- The error due to policy update (characterized by $\beta$) can be made arbitrarily small by choosing $\beta$ large. However, choosing a large $\beta$ may increase $C$ due to the fact that $C$ is a function of the mixing time of each policy, which is affected by the choice of $\beta$. However, $C$ cannot be arbitrarily large due to Assumption (a).

- The error due to the initial condition decays exponentially fast but the rate of decay can be slow because $\gamma$ can be very small. In this regard, it is interesting to compare our work with the results in [AYBB+19, LYAYS21]. We do so in the next subsection.

## 4.2 Comparison with [AYBB$^+$19, LYAYS21]

Our model for approximate policy iteration is intended to be a general framework to study policy-based RL algorithms. A specific class of such algorithms has been studied in [AYBB$^+$19, LYAYS21], where regret bounds are presented for a mirror descent like algorithm referred to as POLITEX. Although our model is intended for offline algorithms, it is possible to adapt the techniques in [AYBB$^+$19, LYAYS21] to obtain regret bounds for our model as well.

Let $r(s_t, a_t)$ be the reward obtained at time $t$. Let $K$ be the total number of time units for which Algorithm 3 is executed. Hence $K = T\tau$, where $\tau$ is the length of the sample trajectory for every iteration of policy evaluation. The regret, defined as $R(K) = \sum_{t=1}^{K} (J^* - r(s_t, a_t))$ can be further decomposed as:

$$R(K) = \sum_{t=1}^{K} (J^* - J_{\mu_t}) + (J_{\mu_t} - r(s_t, a_t)). \tag{14}$$

The first term i.e., $\sum_{t=1}^{K} (J^* - J_{\mu_t})$ is referred to as the pseudo regret $R_{\text{PS}}(K)$. The second term is essentially bounding the average value from its estimate and can be bounded using a martingale analysis identical to [AYBB$^+$19, LYAYS21]. Here, we focus on obtaining an upper bound on the pseudo regret with mirror descent update and TD learning. The pseudo regret can be further decomposed as:

$$R_{\text{PS}}(K) = \tau \sum_{t=1}^{T} (J^* - J_{\mu_t}). $$

For the mirror descent policy update in Corollary 4.5, we get the following regret bound

$$R_{\text{PS}}(K) \le \tau \sum_{t=1}^{T} (1-\gamma)^T c_0 + \frac{1+\gamma}{\gamma\beta} \log\left(\frac{1}{\omega}\right) + \frac{2}{\gamma}\left(\bar{\delta}_0 + \frac{c_3}{\sqrt{\tau}}\right). \tag{15}$$

Let $\beta = \sqrt{\tau}$, then the above regret can be expressed as:

$$R_{\text{PS}}(K) \le \frac{1}{\gamma}\left(\tau c_0 + 2K\bar{\delta}_0 + \frac{Kc_5}{\sqrt{\tau}}\right). $$

Optimizing for the regret yields $\tau = O(K^{\frac{2}{3}})$ and the following pseudo regret upper bound,

$$R_{\text{PS}}(K) \le \frac{1}{\gamma}\left(2K\bar{\delta}_0 + c_6 K^{\frac{2}{3}}\right). \tag{16}$$

We now compare our regret bounds to the ones in [AYBB$^+$19, LYAYS21]. Since all of the bounds will have an inevitable error $(\bar{\delta}_0)$ due to function approximation, the comparison is with respect to the other terms in the regret bound.

- We get a better order-wise bound on the regret compare to [AYBB$^+$19] which obtains a bound of $O(K^{3/4})$. However, they have better constants which are in terms of mixing-time coefficients rather than in terms of $\gamma$.
- [LYAYS21] gets a better regret bound $(O(\sqrt{K}))$ than us.

The results in [AYBB$^+$19, LYAYS21] use Monte Carlo policy evaluation, which allows for a closed-form expression for the value function estimate. The structure of the closed-form expressions allows them to exploit the fact that the policy varies slowly from one iteration to the next (which is specific to the mirror descent policy update) and hence $\gamma$ does not appear in their analysis. However, the Monte Carlo policy evaluation algorithms use matrix inversion which is often too expensive to implement in practice unlike the TD learning algorithm in our analysis which is widely used. Nevertheless, the ideas in [AYBB$^+$19, LYAYS21] provide a motivation to investigate whether one can extend their analysis to more general policy evaluation and policy improvement algorithms. This is an interesting topic for future research.

# 5 Conclusions

We present a general framework for analyzing policy-based reinforcement learning algorithms for average-reward MDPs. Our main contribution is to obtain performance bounds without using the natural source of contraction available in discounted reward formulations. We apply our general framework to several well-known RL algorithms to obtain finite-time error bounds on the average reward. We conclude with a comparative regret analysis and an outline of possible future work.

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
