# Appendix: Performance Bounds for Policy-Based Average Reward Reinforcement Learning Algorithms

## 1 Discussion on Assumption 3.1

### 1.1 Assumption 3.1(a)

In order to ensure every policy induces an irreducible Markov chain, we modify the MDP where at every time step with probability $\epsilon$, an action is chosen from the set of all possible actions with equal probability. Simultaneously, with probability $1 - \epsilon$, we choose an action dictated by some policy. Let the transition kernel of the MDP associated with policy $\mu$ be denoted as $\mathbb{P}_\mu$. We denote the modified kernel by $\widehat{\mathbb{P}}_\mu$, where the transition probability from state $i$ to $j$ is given by:

$$\widehat{\mathbb{P}}_\mu(j|i) = (1-\epsilon)\mathbb{P}_\mu(j|i) + \epsilon\left(\frac{1}{|\mathcal{A}|}\sum_{a\in\mathcal{A}}\mathbb{P}(j|i,a)\right)$$

Let $\mathbb{P}_\rho(j|i) = \frac{1}{|\mathcal{A}|}\sum_{a\in\mathcal{A}}\mathbb{P}(j|i,a)$, where $\rho$ represents randomized policy, that is $\rho(a|i) = \frac{1}{|\mathcal{A}|}$. Let $(J_\mu, V_\mu)$ be the average reward and state value function vector associated with the policy $\mu$. Then they the satisfy the following Bellman Equation:

$$J_\mu + V_\mu = r_\mu + \mathbb{P}_\mu V_\mu,$$

where $r_\mu(i) = \sum_{a\in\mathcal{A}}\mu(a|i)r(i,a)$. Similarly $(\widehat{J}_\mu, \widehat{V}_\mu)$ be the average reward and state value function vector that satisfy the following Bellman Equation,

$$\widehat{J}_\mu + \widehat{V}_\mu = \widehat{r}_\mu + \widehat{\mathbb{P}}_\mu\widehat{V}_\mu$$

where $\widehat{r}_\mu = (1-\epsilon)r_\mu + \epsilon r_\rho$. Since $\widehat{\mathbb{P}}_\mu = (1-\epsilon)\mathbb{P}_\mu + \epsilon\mathbb{P}_\rho$, the above equation can be rewritten as:

$$\widehat{J}_\mu = (1-\epsilon)\left(r_\mu + \mathbb{P}_\mu\widehat{V}_\mu - \widehat{V}_\mu\right) + \epsilon\left(r_\rho + \mathbb{P}_\rho\widehat{V}_\mu - \widehat{V}_\mu\right)$$

Multiplying the above equation by the vector $\mathbb{P}_\mu^*$, which is the invariant distribution over the state space due to policy $\mu$, we obtain,

$$\widehat{J}_\mu = (1-\epsilon)\left(\mathbb{P}_\mu^* r_\mu\right) + \epsilon\mathbb{P}_\mu^*\left(r_\rho + \mathbb{P}_\rho\widehat{V}_\mu - \widehat{V}_\mu\right)$$

since $\mathbb{P}_\mu^*\mathbb{P}_\mu = \mathbb{P}_\mu^*$. We also know that $J_\mu = \mathbb{P}_\mu^* r_\mu$.. Hence we obtain,

$$\widehat{J}_\mu = (1-\epsilon)J_\mu + \epsilon\mathbb{P}_\mu^*\left(r_\rho + \mathbb{P}_\rho\widehat{V}_\mu - \widehat{V}_\mu\right)$$

Therefor we obtain the following result,

$$J_\mu - \widehat{J}_\mu = \epsilon\left(J_\mu - \mathbb{P}_\mu^*\left(r_\rho + \mathbb{P}_\rho\widehat{V}_\mu - \widehat{V}_\mu\right)\right)$$

Since the rewards are bounded, so are $J_\mu, r_\rho$ and $\mathbb{P}_\rho\widehat{V}_\mu - \widehat{V}_\mu$ for all $\mu$. Hence we obtain,

$$J_\mu - \widehat{J}_\mu = O(\epsilon)$$

The difference in the average reward associated with the original MDP and the modified MDP vary by $O(\epsilon)$. Let the optimal policy corresponding to original MDP be $\mu^*$ and the optimal policy corresponding to modified MDP be $\widehat{\mu}^*$. Then,

$$J_{\mu^*} - \widehat{J}_{\widehat{\mu}^*} = J_{\mu^*} - \widehat{J}_{\mu^*} + \widehat{J}_{\mu^*} - \widehat{J}_{\widehat{\mu}^*}$$

Since $\widehat{\mu}^*$ is the optimizing policy for the modified MDP, we have $\widehat{J}_{\mu^*} - \widehat{J}_{\widehat{\mu}^*} \leq 0$. Hence, we obtain,

$$J_{\mu^*} - \widehat{J}_{\widehat{\mu}^*} \leq J_{\mu^*} - \widehat{J}_{\mu^*} = O(\epsilon).$$

Similarly,

$$J_{\mu^*} - \widehat{J}_{\widehat{\mu}^*} = J_{\mu^*} - J_{\widehat{\mu}^*} + J_{\widehat{\mu}^*} - \widehat{J}_{\widehat{\mu}^*}$$

Since $\mu^*$ is the optimizing policy for the original MDP, we have $J_{\mu^*} - J_{\widehat{\mu}^*} \geq 0$. Hence we obtain,

$$J_{\mu^*} - \widehat{J}_{\widehat{\mu}^*} \geq J_{\widehat{\mu}^*} - \widehat{J}_{\widehat{\mu}^*} = O(\epsilon).$$

Thus the optimal average reward of the original MDP and modified MDP differ by $O(\epsilon)$. That is,

$$|J_{\mu^*} - \widehat{J}_{\widehat{\mu}^*}| \leq O(\epsilon)$$

## 1.2 Assumption 3.1 (b)

To ensure Assumption 3.1 (b) is satisfied, an aperiodicity transformation can be implemented. Under this transformation, the transition probabilities of the original MDP are modified such that the probability of staying in any state under any policy is non-zero. In order to compensate for the change in transition kernel, the single step rewards are analogously modified such that the Bellman equation corresponding to the original and transformed dynamics are scaled versions of one another. This thus ensures that the optimality of a policy remains the same irrespective of this transformation, along with yielding quantifiable convergence properties. Mathematically, this transformation is described below.

**Definition 1.1** (Aperiodicity Transformation). Let $\kappa \in (0,1)$. For every policy $\mu \in \Pi$ and for all states $i, j \in \mathcal{S}$, consider the following transformation:

$$\widehat{\mathbb{P}}_\mu(i|i) = \kappa + (1-\kappa)\mathbb{P}_\mu(i|i) \tag{1}$$

$$\widehat{\mathbb{P}}_\mu(j|i) = (1-\kappa)\mathbb{P}_\mu(j|i), \qquad j \neq i \tag{2}$$

$$\widehat{r}(i, \mu(i)) = (1-\kappa)r(i, \mu(i)) \tag{3}$$

**Theorem 1.2.** *Given the transformation in Equation (1)-Equation (3), for every $\mu \in \Pi$, let $(J_\mu, h_\mu)$ and $(\widehat{J}_\mu, \widehat{h}_\mu)$ be the solution to the Bellman Equation (Equation 2) corresponding to the original MDP $(\mathbb{P}_\mu, r_\mu)$ and the transformed MDP $(\widehat{\mathbb{P}}_\mu, \widehat{r}_\mu)$. Then,*

$$\left(\widehat{J}_\mu, \widehat{h}_\mu\right) = ((1-\kappa)J_\mu, h_\mu)$$

*Proof.* The proof of this theorem can be found in [Sch71]. $\qquad\square$

*Remark* 1.3. Due to this bijective relationship between the original and the transformed problem, it suffices to solve the transformed problem. Given that the trajectory used for algorithms such as TD Learning corresponds to the original system, such a transformation necessitates a mild change in how samples from a given policy are utilized in TD learning. More specifically, for each (state, action) sample, with probability $\kappa$, we have to add the same sample to the data set for the next time instant. With probability $1 - \kappa$ choose the next state in the trajectory. Hence, this transformation is easy to incorporate in RL algorithms.

## 2 Proof of Theorem 3.3

Prior to presenting the proof, we define

$$u_k = \max_i (\mathsf{T}h_k - h_k)(i), \tag{4}$$

$$l_k = \min_i (\mathsf{T}h_k - h_k)(i).$$

A key lemma in the proof of convergence of approximate policy iteration in the context of average reward is:

**Lemma 2.1.** *Let $J^*$ be the optimal average reward associated with the transformed MDP. For all $k \in \mathbb{N}$:*

$$l_k - \epsilon \leq J_{\mu_{k+1}} \leq J^* \leq u_k$$

*Proof.* From definition,

$$l_k \mathbf{1} \leq \mathsf{T} h_k - h_k$$

Since $\|\mathsf{T} h_k - \mathsf{T}_{\mu_{k+1}} h_k\|_\infty \leq \epsilon$,

$$l_k \mathbf{1} \leq \mathsf{T}_{\mu_{k+1}} h_k - h_k + \epsilon \mathbf{1}$$
$$= r_{\mu_{k+1}} + \mathbb{P}_{\mu_{k+1}} h_k - h_k + \epsilon \mathbf{1}$$
$$\mathbb{P}^*_{\mu_{k+1}} l_k \mathbf{1} \leq \mathbb{P}^*_{\mu_{k+1}} r_{\mu_{k+1}} + \mathbb{P}^*_{\mu_{k+1}} \mathbb{P}_{\mu_{k+1}} h_k - \mathbb{P}^*_{\mu_{k+1}} h_k + \mathbb{P}^*_{\mu_{k+1}} \epsilon \mathbf{1}$$

where $\mathbb{P}^*_{\mu_{k+1}}$ is a matrix whose rows are identical and are the invariant distribution associated with the probability transition kernel $\mathbb{P}_{\mu_{k+1}}$. Since $\mathbb{P}^*_{\mu_{k+1}} \mathbb{P}_{\mu_{k+1}} = \mathbb{P}^*_{\mu_{k+1}}$,

$$l_k \mathbf{1} \leq \mathbb{P}^*_{\mu_{k+1}} r_{\mu_{k+1}} + \epsilon \mathbf{1}$$
$$l_k \mathbf{1} - \epsilon \mathbf{1} \leq J_{\mu_{k+1}} \mathbf{1}$$
$$l_k - \epsilon \leq J_{\mu_{k+1}}$$

From definition,

$$J^* = \max_{\mu \in \Pi} J_\mu \geq J_{\mu_{k+1}}$$

Let $\mu^*$ be the policy corresponding to the optimal average reward $J^*$ and value function $h^*$, ie.,

$$J^* + h^* = \max_\mu r_\mu + \mathbb{P}_\mu h^*$$
$$= r_{\mu^*} + \mathbb{P}_{\mu^*} h^*$$

Then,

$$u_k \mathbf{1} \geq \mathsf{T} h_k - h_k$$
$$\overset{(a)}{\geq} \mathsf{T}_{\mu^*} h_k - h_k$$
$$= r_{\mu^*} + \mathbb{P}_{\mu^*} h_k - h_k$$
$$\mathbb{P}^*_{\mu^*} u_k \mathbf{1} \geq \mathbb{P}^*_{\mu^*} r_{\mu^*} + \mathbb{P}^*_{\mu^*} \mathbb{P}_{\mu^*} h_k - \mathbb{P}^*_{\mu^*} h_k$$
$$u_k \mathbf{1} \geq \mathbb{P}^*_{\mu^*} r_{\mu^*}$$
$$u_k \geq J^*$$

where $(a)$ is due to the fact that $\mathsf{T}$ is the maximising Bellman operator.
Hence $\forall k \in \mathbb{N}$,

$$l_k - \epsilon \leq J_{\mu_{k+1}} \leq J^* \leq u_k$$

$\square$

Without any approximation, the above lemma indicates that there exists a state whose value function changes by an amount less than the average reward associated with the optimizing policy at that iteration, and also that there exists a state whose value function increases by an amount larger than the optimal average reward.

The proof proceeds by showing a geometric convergence rate for $l_k$. More precisely, the following lemma is proved.

**Lemma 2.2.** *For all $k \in \mathbb{N}$, it is true that:*

$$(J^* - l_k) \leq (1 - \gamma)(J^* - l_{k-1}) + \epsilon + 2\delta$$

*Proof.* Define:

$$g_k(i) = (\mathsf{T} h_k - h_k)(i)$$
$$g_k(i) \geq (\mathsf{T}_{\mu_k} h_k - h_k)(i)$$

Since $\mathsf{T}$ is the maximizing Bellman operator.
We know that:

1. $\|h_k - h_{\mu_k}\|_\infty \leq \delta$

2. For all constant $p \in \mathbb{R}$:

$$\mathsf{T}_{\mu_k}(h_k + p\mathbf{1}) = r_{\mu_k} + \mathbb{P}_{\mu_k}(h_k + p\mathbf{1})$$
$$= r_{\mu_k} + \mathbb{P}_{\mu_k}h_k + p\mathbf{1}$$
$$= \mathsf{T}_{\mu_k}h_k + p\mathbf{1}$$

We thus obtain,

$$g_k(i) \geq \mathsf{T}_{\mu_k}(h_{\mu_k} - \delta) - (h_{\mu_k} + \delta) \tag{5}$$
$$\geq \left(\mathsf{T}_{\mu_k}h_{\mu_k} - h_{\mu_k}\right)(i) - 2\delta \tag{6}$$

Recall that $h_{\mu_k} = \lim_{m\to\infty} \widetilde{T}^m_{\mu_{k+1}} h_k$

$$\widetilde{\mathsf{T}}^m_{\mu_k}h_{k-1} = \widetilde{\mathsf{T}}^{m-1}_{\mu_k}\widetilde{\mathsf{T}}_{\mu_k}h_{k-1}$$
$$= \widetilde{\mathsf{T}}^{m-1}_{\mu_k}\left(r_{\mu_k} + \mathbb{P}_{\mu_k}h_{k-1} - r_{\mu_k}(x^*)\mathbf{1} - (\mathbb{P}_{\mu_k}h_{k-1})(x^*)\mathbf{1}\right)$$
$$= \widetilde{\mathsf{T}}^{m-1}_{\mu_k}\left(\mathsf{T}_{\mu_k}h_{k-1} - (\mathsf{T}_{\mu_k}h_{k-1})(x^*)\mathbf{1}\right)$$
$$= \widetilde{\mathsf{T}}^{m-2}_{\mu_k}\widetilde{\mathsf{T}}_{\mu_k}\left(\mathsf{T}_{\mu_k}h_{k-1} - (\mathsf{T}_{\mu_k}h_{k-1})(x^*)\mathbf{1}\right)$$
$$= \widetilde{\mathsf{T}}^{m-2}_{\mu_k}\left(r_{\mu_k} + \mathbb{P}_{\mu_k}\left(\mathsf{T}_{\mu_k}h_{k-1} - \mathsf{T}_{\mu_k}h_{k-1}(x^*)\right) - r_{\mu_k}(x^*)\mathbf{1} - (\mathbb{P}_{\mu_k}\left(\mathsf{T}_{\mu_k}h_{k-1} - \mathsf{T}_{\mu_k}h_{k-1}(x^*)\right))(x^*)\mathbf{1}\right)$$
$$= \widetilde{\mathsf{T}}^{m-2}_{\mu_k}\left(r_{\mu_k} + \mathbb{P}_{\mu_k}\left(\mathsf{T}_{\mu_k}h_{k-1}\right) - \mathsf{T}_{\mu_k}h_{k-1}(x^*) - r_{\mu_k}(x^*)\mathbf{1} - \mathbb{P}_{\mu_k}\left(\mathsf{T}_{\mu_k}h_{k-1}\right)(x^*)\mathbf{1} + \mathsf{T}_{\mu_k}h_{k-1}(x^*)\right)$$
$$= \widetilde{\mathsf{T}}^{m-2}_{\mu_k}\left(\mathsf{T}^2_{\mu_k}h_{k-1} - \mathsf{T}^2_{\mu_k}h_{k-1}(x^*)\mathbf{1}\right)$$

Iterating, we thus obtain,

$$h_{\mu_k} = \lim_{m\to\infty} \mathsf{T}^m_{\mu_k}h_{k-1} - \mathsf{T}^m_{\mu_k}h_{k-1}(x^*)\mathbf{1}$$

Substituting back in Equation (5),

$$g_k(i) \geq \left(\mathsf{T}_{\mu_k}\left(\lim_{m\to\infty}\mathsf{T}^m_{\mu_k}h_{k-1} - \mathsf{T}^m_{\mu_k}h_{k-1}(x^*)\mathbf{1}\right) - \lim_{m\to\infty}\mathsf{T}^m_{\mu_k}h_{k-1} - \mathsf{T}^m_{\mu_k}h_{k-1}(x^*)\mathbf{1}\right)(i) - 2\delta$$
$$= \left(\mathsf{T}_{\mu_k}\left(\lim_{m\to\infty}\mathsf{T}^m_{\mu_k}h_{k-1}\right) - \lim_{m\to\infty}\mathsf{T}^m_{\mu_k}h_{k-1}(x^*)\mathbf{1} - \lim_{m\to\infty}\mathsf{T}^m_{\mu_k}h_{k-1} - \lim_{m\to\infty}\mathsf{T}^m_{\mu_k}h_{k-1}(x^*)\mathbf{1}\right)(i) - 2\delta$$
$$= \left(\mathsf{T}_{\mu_k}\left(\lim_{m\to\infty}\mathsf{T}^m_{\mu_k}h_{k-1}\right) - \lim_{m\to\infty}\mathsf{T}^m_{\mu_k}h_{k-1}\right)(i) - 2\delta$$

Since $\mathsf{T}_{\mu_k}$ is a continuous operator, it is possible to take the limit outside. We thus obtain,

$$g_k(i) \geq \lim_{m\to\infty}\left(\mathsf{T}^{m+1}_{\mu_k}h_{k-1} - \mathsf{T}^m_{\mu_k}h_{k-1}\right)(i) - 2\delta$$
$$= \lim_{m\to\infty}\left(\mathsf{T}^m_{\mu_k}\mathsf{T}_{\mu_k}h_{k-1} - \mathsf{T}^m_{\mu_k}h_{k-1}\right)(i) - 2\delta$$
$$= \lim_{m\to\infty}\left(\mathbb{P}^m_{\mu_k}\left(\mathsf{T}_{\mu_k}h_{k-1} - h_{k-1}\right)\right)(i) - 2\delta$$
$$= \mathbb{P}^*_{\mu_k}\left(\mathsf{T}_{\mu_k}h_{k-1} - h_{k-1}\right)(i) - 2\delta$$

Since $\|\mathsf{T}_{\mu_k}h_{k-1} - Th_{k-1}\|_\infty \leq \epsilon$,

$$g_k(i) \geq \mathbb{P}^*_{\mu_k}\left(Th_{k-1} - h_{k-1}\right)(i) - 2\delta - \epsilon$$
$$= \left(\mathbb{P}^*_{\mu_k}g_{k-1}\right)(i) - 2\delta - \epsilon$$

Note that from Lemma 3.2 in the mainpaper, we know that

$$\min_{i,j\in\mathcal{S}}\mathbb{P}^*_{\mu_k}(j|i) > \gamma > 0$$

Also, since every policy is assumed to induce an irreducible Markov Process, we have a limiting distribution with all positive entries. We thus obtain the following crucial relationship,

$$g_k(i) \geq (1-\gamma)l_{k-1} + \gamma u_{k-1} - 2\delta - \epsilon \tag{7}$$

Since the above inequation is true for all states $i$, it has to also be true for $\text{argmin}_i \left( \mathsf{T}h_k - h_k \right)(i)$. Hence, we obtain,

$$l_k \geq (1 - \gamma)l_{k-1} + \gamma u_{k-1} - 2\delta - \epsilon$$

Note that from Lemma 2.1, we know that $u_k \geq J^*$ for all $k \in \mathbb{N}$.
Hence we obtain,

$$l_k \geq (1 - \gamma)l_{k-1} + \gamma J^* - 2\delta - \epsilon$$

Upon rearranging we obtain the result, that is,

$$(J^* - l_k) \leq (1 - \gamma)(J^* - l_{k-1}) + 2\delta + \epsilon$$

$\square$

We now present the proof of Theorem 3.3.

From Lemma 2.2, we thus have,

$$(J^* - l_k) \leq (1 - \gamma)(J^* - l_{k-1}) + 2\delta + \epsilon \tag{8}$$

However, we know that,

$$l_k - \epsilon \leq J^*$$

In order to iterate Equation (8), need to ensure the terms are non-negative. Rearranging the terms thus yields,

$$(J^* - l_k + \epsilon) \leq (1 - \gamma)(J^* - l_{k-1} + \epsilon) + 2\delta + (1 + \gamma)\epsilon$$

Let

$$\omega = 2\delta + (1 + \gamma)\epsilon$$

and

$$a_k = J^* - l_k + \epsilon.$$

Then upon iterating, we obtain,

$$\begin{aligned}
a_k &\leq (1 - \gamma)a_{k-1} + \omega \\
&\leq (1 - \gamma)((1 - \gamma)a_{k-2} + \omega) + \omega \\
&\leq \omega \frac{1 - (1 - \gamma)^k}{\gamma} + (1 - \gamma)^k a_0
\end{aligned}$$

Substituting back, we obtain,

$$(J^* - l_k + \epsilon) \leq \left( \frac{1 - (1 - \gamma)^k}{\gamma} \right)(2\delta + (1 + \gamma)\epsilon) + (1 - \gamma)^k(J^* - l_0 + \epsilon) \tag{9}$$

$$(J^* - l_k) \leq \left( \frac{1 - (1 - \gamma)^k}{\gamma} \right)(2\delta + (1 + \gamma)\epsilon) - \epsilon + (1 - \gamma)^k(J^* - l_0 + \epsilon)$$

Note that from Lemma 2.1 we know that

$$l_k \leq J_{\mu_{k+1}} + \epsilon \leq J^* + \epsilon$$

Hence we get,

$$\left( J^* - J_{\mu_{k+1}} \right) \leq (J^* - l_k + \epsilon)$$

Thus we obtain,

$$\left( J^* - J_{\mu_{k+1}} \right) \leq \left( \frac{1 - (1 - \gamma)^k}{\gamma} \right)(2\delta + (1 + \gamma)\epsilon) + (1 - \gamma)^k(J^* - l_0 + \epsilon)$$

Theorem 3.3 presents an upper bound on the error in terms of the average reward. Recall that, for the optimal relative value function $h$, the following relationship holds: $Th - h = J^*\mathbf{1}$. Thus, it is also interesting to understand how $Th_k - h_k$ behaves under approximate policy iteration. The following proposition characterizes the behavior of this term.

**Proposition 2.3.** *The iterates generated from approximate policy iteration algorithm 1 satisfy the following bound:*

$$(u_{k-1} - l_{k-1}) \leq \underbrace{\left( \frac{1 - (1-\gamma)^k}{\gamma^2} \right)(2\delta + (1+\gamma)\epsilon) + \frac{2\delta + \epsilon}{\gamma}}_{approximation\ error} + \underbrace{\frac{(1-\gamma)^k(J^* - l_0 + \epsilon)}{\gamma}}_{initial\ condition\ error},$$

*where* $u_k, l_k$ *are defined in Equation* (4).

*Proof.* It is known from Equation (7) that,

$$g_k(i) \geq (1-\gamma)l_{k-1} + \gamma u_{k-1} - 2\delta - \epsilon$$

This further yields,

$$l_k \geq (1-\gamma)l_{k-1} + \gamma u_{k-1} - 2\delta - \epsilon$$

From Lemma 2.1, we know that,

$$l_k \leq J^* + \epsilon$$
$$J^* + \epsilon \geq (1-\gamma)l_{k-1} + \gamma u_{k-1} - 2\delta - \epsilon$$
$$J^* - l_k + \epsilon \geq \gamma(u_{k-1} - l_{k-1}) - 2\delta - \epsilon$$
$$\gamma(u_{k-1} - l_{k-1}) \leq (J^* - l_k + \epsilon) + 2\delta + \epsilon$$

From Equation (9), we thus obtain,

$$\gamma(u_{k-1} - l_{k-1}) \leq \left( \frac{1 - (1-\gamma)^k}{\gamma} \right)(2\delta + (1+\gamma)\epsilon) + (1-\gamma)^k(J^* - l_0 + \epsilon) + 2\delta + \epsilon$$

We thus obtain the result in Proposition 3.11,

$$(u_{k-1} - l_{k-1}) \leq \left( \frac{1 - (1-\gamma)^k}{\gamma^2} \right)(2\delta + (1+\gamma)\epsilon) + \frac{2\delta + \epsilon}{\gamma} + \frac{(1-\gamma)^k(J^* - l_0 + \epsilon)}{\gamma}$$

$\square$

**Corollary 2.4.** *The asymptotic behavior of the relative value function iterates is given by*

$$\limsup_{k \to \infty} (u_k - l_k) \leq \frac{\epsilon(1 + 2\gamma) + 2\delta(1 + \gamma)}{\gamma^2}$$

**Comment of the Novelty of our Proof Technique:** As mentioned in the main body of the paper, our proof is inspired by the proof of convergence of modified policy iteration in [VdW80]. However, since our algorithm is quite different from modified policy iteration due to the presence of errors in each step of the algorithm, special care is needed to obtain useful performance bounds. Specifically, the impact of the errors at each step have to be carefully bounded to ensure that the performance bounds do not blow up to infinity as it does when we obtain a similar result for the discounted-reward case and let the discount factor go to one.

## 3 Proofs from Section 4

Algorithm 2 and its analysis, adapted to the context of $Q$ function with time dependent approximation errors is presented in this section. Most RL algorithms use the state-action relative value function $Q$ instead of the relative state value function $h$ to evaluate a policy. The corresponding Bellman Equation in terms of $Q$ associated with any state-action pair $(s, a)$ is given by

$$J_\mu + Q_\mu(s, a) = r(s, a) + (\mathbb{Q}_\mu Q_\mu)(s, a)$$

where $\mathbb{Q}_\mu(s', a'|s, a) = \mu(a'|s')\mathbb{P}(s'|s, a)$. Since we are interested in randomized policies for exploration reasons, the irreducibility assumptions imposed on $\mathbb{P}_\mu$ also hold true for the transition kernel $\mathbb{Q}_\mu$. The state relative value function $h_\mu$ and state-action relative value function $Q_\mu$ are related as $h_\mu(s) = \sum_a \mu(a|s)Q_\mu(s, a)$. Consider the following similar definitions of the Bellman operators corresponding to the state-action value function:

$$(\mathsf{T}_\mu^{\mathsf{Q}}Q)(s, a) = r(s, a) + (\mathbb{Q}_\mu Q)(s, a)$$

and

$$(\mathsf{T}^Q Q)(s,a) = r(s,a) + \max_{\mu \in \Pi} (\mathbb{Q}_\mu Q)(s,a).$$

Let $(s^*, a^*)$ represent some fixed state. Then the relative Bellman operator, relative to $(s^*, a^*)$ is defined as:

$$\left(\widetilde{\mathsf{T}}^Q_\mu Q\right)(s,a) = r(s,a) + (\mathbb{Q}_\mu Q)(s,a) - r(s^*, a^*) - (\mathbb{Q}_\mu Q)(s^*, a^*).$$

The algorithm, thus adapted to state action relative value function is given below:

---

**Algorithm 1** Approximate Policy Iteration: Average Reward

---
1: Require $Q_0 \in \mathbb{R}^n$
2: **for** $k = 0, 1, 2, \dots$ **do**
3:     1. Compute $\mu_{k+1} \in \Pi$ such that $\|\mathsf{T}Q_k - \mathsf{T}_{\mu_{k+1}} Q_k\|_\infty \leq \epsilon_k$
4:     2. Compute $Q_{k+1}$ such that $\|Q_{k+1} - h_{\mu_{k+1}}\|_\infty \leq \delta_{k+1}$
5:         where $Q_{\mu_{k+1}} = \lim_{m \to \infty} \left(\widetilde{\mathsf{T}}^Q_{\mu_{k+1}}\right)^m Q_k$
6: **end for**

---

Define

$$g_k(s,a) = \left(\mathsf{T}^Q Q_k - Q_k\right)(s,a), \qquad \forall (s,a) \in \mathcal{S} \times \mathcal{A},$$

and set

$$l_k = \min_{(s,a) \in \mathcal{S} \times \mathcal{A}} g_k(s,a), \qquad u_k = \max_{(s,a) \in \mathcal{S} \times \mathcal{A}} g_k(s,a).$$

We prove the convergence in three parts:

1. We prove that $l_k - \epsilon_k \leq J_{\mu_{k+1}} \leq J^* \leq u_k$.
2. We use the result from 1 to prove Lemma 4.1.
3. We iteratively employ the result in Lemma 4.1 to prove Proposition 4.2.

### 3.1 Proof of part 1

By definition, we have

$$l_k \mathbf{1} \leq \mathsf{T}^Q Q_k - Q_k.$$

Since $\|\mathsf{T}^Q Q_k - \mathsf{T}^Q_{\mu_{k+1}} Q_k\|_\infty \leq \epsilon_k$, it follows that

$$l_k \mathbf{1} \leq \mathsf{T}^Q_{\mu_{k+1}} Q_k - Q_k + \epsilon_k \mathbf{1}$$
$$= r + \mathbb{Q}_{\mu_{k+1}} Q_k - Q_k + \epsilon_k \mathbf{1}.$$

Multiplying by $\mathbb{Q}^*_{\mu_{k+1}}$, we get

$$\mathbb{Q}^*_{\mu_{k+1}} l_k \mathbf{1} \leq \mathbb{Q}^*_{\mu_{k+1}} r + \mathbb{Q}^*_{\mu_{k+1}} \mathbb{Q}_{\mu_{k+1}} Q_k - \mathbb{Q}^*_{\mu_{k+1}} Q_k + \mathbb{Q}^*_{\mu_{k+1}} \epsilon_k \mathbf{1}$$
$$= \mathbb{Q}^*_{\mu_{k+1}} r + \epsilon_k \mathbf{1} \qquad (10)$$

where $\mathbb{Q}^*_{\mu_{k+1}}$ is a matrix whose rows are identical and are the invariant distribution associated with the probability transition kernel $\mathbb{Q}_{\mu_{k+1}}$. Note that $\mathbb{Q}^*_{\mu_{k+1}} \mathbf{1} = \mathbf{1}$, and that $\mathbb{Q}^*_{\mu_{k+1}} \mathbb{Q}_{\mu_{k+1}} = \mathbb{Q}^*_{\mu_{k+1}}$.

Consider the Bellman Equation corresponding to the state-action relative value function associated with the policy $\mu_{k+1}$:

$$J_{\mu_{k+1}} + Q_{\mu_{k+1}} = r + \mathbb{Q}_{\mu_{k+1}} Q_{\mu_{k+1}}$$

It follows that

$$\mathbb{Q}^*_{\mu_{k+1}} J_{\mu_{k+1}} + \mathbb{Q}^*_{\mu_{k+1}} Q_{\mu_{k+1}} = \mathbb{Q}^*_{\mu_{k+1}} r + \mathbb{Q}^*_{\mu_{k+1}} \mathbb{Q}_{\mu_{k+1}} Q_{\mu_{k+1}}.$$

Since $\mathbb{Q}^*_{\mu_{k+1}} \mathbb{Q}_{\mu_{k+1}} = \mathbb{Q}^*_{\mu_{k+1}}$, we have that

$$J_{\mu_{k+1}} \mathbf{1} = \mathbb{Q}^*_{\mu_{k+1}} r.$$

Hence, Equation (10) yields

$$l_k \le J_{\mu_{k+1}} + \epsilon_k.$$

Equivalently, we have

$$l_k - \epsilon_k \le J_{\mu_{k+1}}.$$

From definition,

$$J^* = \max_{\mu \in \Pi} J_\mu \ge J_{\mu_{k+1}}.$$

Let $\mu^*$ be the policy corresponding to the optimal average reward $J^*$. Let $Q^*$ denote the state-action relative value function associated with the policy $\mu^*$. Note that $(J^*, Q^*)$ is the solution of the Bellman optimality equation, i.e.,

$$J^* + Q^* = r + \max_{\mu \in \Pi} \mathbb{Q}_\mu Q^*$$
$$= r + \mathbb{Q}_{\mu^*} Q^*.$$

Then,

$$u_k \mathbf{1} \ge \mathsf{T}^{\mathsf{Q}} Q_k - Q_k$$
$$\overset{(a)}{\ge} \mathsf{T}^{\mathsf{Q}}_{\mu^*} Q_k - Q_k$$
$$= r + \mathbb{Q}_{\mu^*} Q_k - Q_k,$$

where (a) is due to the fact that $\mathsf{T}^{\mathsf{Q}}$ is the Bellman optimality operator. Therefore, we have

$$\mathbb{Q}^*_{\mu^*} u_k \mathbf{1} \ge \mathbb{Q}^*_{\mu^*} r + \mathbb{Q}^*_{\mu^*} \mathbb{Q}_{\mu^*} Q_k - \mathbb{Q}^*_{\mu^*} Q_k$$

Equivalently,

$$u_k \mathbf{1} \ge \mathbb{Q}^*_{\mu^*} r$$

Therefore, we conclude that

$$u_k \ge J^*$$

Hence, for all $k \in \mathbb{N}$,

$$l_k - \epsilon_k \le J_{\mu_{k+1}} \le J^* \le u_k \tag{11}$$

## 3.2 Proof of Lemma 4.1

Recall that

$$g_k(s, a) = \left( \mathsf{T}^{\mathsf{Q}} Q_k - Q_k \right)(s, a) \ge \left( \mathsf{T}^{\mathsf{Q}}_{\mu_k} Q_k - Q_k \right)(s, a),$$

where the inequality follows by the fact that $\mathsf{T}^{\mathsf{Q}}$ is the Bellman optimality operator. Note that the Bellman operator $\mathsf{T}^{\mathsf{Q}}_{\mu_k}$ is shift-invariant, i.e., for all $p \in \mathbb{R}$:

$$\mathsf{T}^{\mathsf{Q}}_{\mu_k} (Q_k + p\mathbf{1}) = r + \mathbb{Q}_{\mu_k} (Q_k + p\mathbf{1})$$
$$= r + \mathbb{Q}_{\mu_k} Q_k + p\mathbf{1}$$
$$= \mathsf{T}^{\mathsf{Q}}_{\mu_k} Q_k + p\mathbf{1}$$

Recall that $\|Q_k - Q_{\mu_k}\|_\infty \le \delta_k$. Hence, we have

$$g_k(s, a) \ge \mathsf{T}^{\mathsf{Q}}_{\mu_k} (Q_{\mu_k} - \delta_k \mathbf{1})(s, a) - (Q_{\mu_k}(s, a) + \delta_k)$$
$$\ge \left( \mathsf{T}^{\mathsf{Q}}_{\mu_k} Q_{\mu_k} - Q_{\mu_k} \right)(s, a) - 2\delta_k \tag{12}$$

Recall that $Q_{\mu_k} = \lim_{m\to\infty} \left(\widetilde{\mathsf{T}}^{\mathsf{Q}}_{\mu_{k+1}}\right)^m Q_k$. Therefore,

$$
\begin{aligned}
\left(\widetilde{\mathsf{T}}^{\mathsf{Q}}_{\mu_k}\right)^m Q_{k-1} &= \left(\widetilde{\mathsf{T}}^{\mathsf{Q}}_{\mu_k}\right)^{m-1} \left(\widetilde{\mathsf{T}}^{\mathsf{Q}}_{\mu_k}\right) Q_{k-1} \\
&= \left(\widetilde{\mathsf{T}}^{\mathsf{Q}}_{\mu_k}\right)^{m-1} \left(r + \mathbb{Q}_{\mu_k} Q_{k-1} - r(s^*,a^*)\mathbf{1} - \left(\mathbb{Q}_{\mu_k} Q_{k-1}\right)(s^*,a^*)\mathbf{1}\right) \\
&= \left(\widetilde{\mathsf{T}}^{\mathsf{Q}}_{\mu_k}\right)^{m-1} \left(\mathsf{T}^{\mathsf{Q}}_{\mu_k} Q_{k-1} - \left(\mathsf{T}^{\mathsf{Q}}_{\mu_k} Q_{k-1}\right)(s^*,a^*)\mathbf{1}\right) \\
&= \left(\widetilde{\mathsf{T}}^{\mathsf{Q}}_{\mu_k}\right)^{m-2} \left(\widetilde{\mathsf{T}}^{\mathsf{Q}}_{\mu_k}\right) \left(\mathsf{T}^{\mathsf{Q}}_{\mu_k} Q_{k-1} - \left(\mathsf{T}^{\mathsf{Q}}_{\mu_k} Q_{k-1}\right)(s^*,a^*)\mathbf{1}\right) \\
&= \left(\widetilde{\mathsf{T}}^{\mathsf{Q}}_{\mu_k}\right)^{m-2} \Big(r + \mathbb{Q}_{\mu_k} \left(\mathsf{T}^{\mathsf{Q}}_{\mu_k} Q_{k-1} - \left(\mathsf{T}^{\mathsf{Q}}_{\mu_k} Q_{k-1}\right)(s^*,a^*)\mathbf{1}\right) \\
&\qquad\qquad - r(s^*,a^*)\mathbf{1} - \left(\mathbb{Q}_{\mu_k} \left(\mathsf{T}^{\mathsf{Q}}_{\mu_k} Q_{k-1} - \left(\mathsf{T}^{\mathsf{Q}}_{\mu_k} Q_{k-1}\right)(s^*,a^*)\mathbf{1}\right)\right)(s^*,a^*)\mathbf{1}\Big) \\
&= \left(\widetilde{\mathsf{T}}^{\mathsf{Q}}_{\mu_k}\right)^{m-2} \Big(r + \mathbb{Q}_{\mu_k} \left(\mathsf{T}^{\mathsf{Q}}_{\mu_k} Q_{k-1}\right) - \left(\mathsf{T}^{\mathsf{Q}}_{\mu_k} Q_{k-1}\right)(s^*,a^*)\mathbf{1} \\
&\qquad\qquad - r(s^*,a^*)\mathbf{1} - \mathbb{Q}_{\mu_k} \left(\mathsf{T}^{\mathsf{Q}}_{\mu_k} Q_{k-1}\right)(s^*,a^*)\mathbf{1} + \left(\mathsf{T}^{\mathsf{Q}}_{\mu_k} Q_{k-1}\right)(s^*,a^*)\mathbf{1}\Big) \\
&= \left(\widetilde{\mathsf{T}}^{\mathsf{Q}}_{\mu_k}\right)^{m-2} \left(\left(\mathsf{T}^{\mathsf{Q}}_{\mu_k}\right)^2 Q_{k-1} - \left(\mathsf{T}^{\mathsf{Q}}_{\mu_k}\right)^2 Q_{k-1}(s^*,a^*)\mathbf{1}\right)
\end{aligned}
$$

Iterating, we thus obtain,

$$
Q_{\mu_k} = \lim_{m\to\infty} (\mathsf{T}^{\mathsf{Q}}_{\mu_k})^m Q_{k-1} - (\mathsf{T}^{\mathsf{Q}}_{\mu_k})^m Q_{k-1}(s^*,a^*)\mathbf{1}
$$

Substituting back in Equation (12),

$$
\begin{aligned}
g_k(s,a) &\geq \left(\mathsf{T}^{\mathsf{Q}}_{\mu_k}\left(\lim_{m\to\infty}(\mathsf{T}^{\mathsf{Q}}_{\mu_k})^m Q_{k-1} - (\mathsf{T}^{\mathsf{Q}}_{\mu_k})^m Q_{k-1}(s^*,a^*)\mathbf{1}\right)\right. \\
&\qquad\qquad \left. - \lim_{m\to\infty}\left(\mathsf{T}^{\mathsf{Q}}_{\mu_k}\right)^m Q_{k-1} - \left(\mathsf{T}^{\mathsf{Q}}_{\mu_k}\right)^m Q_{k-1}(s^*,a^*)\mathbf{1}\right)(s,a) - 2\delta_k \\
&= \left(\mathsf{T}^{\mathsf{Q}}_{\mu_k}\left(\lim_{m\to\infty}(\mathsf{T}^{\mathsf{Q}}_{\mu_k})^m Q_{k-1}\right) - \lim_{m\to\infty}(\mathsf{T}^{\mathsf{Q}}_{\mu_k})^m Q_{k-1}(s^*,a^*)\mathbf{1}\right. \\
&\qquad\qquad \left. - \lim_{m\to\infty}\left(\mathsf{T}^{\mathsf{Q}}_{\mu_k}\right)^m Q_{k-1} - \lim_{m\to\infty}\left(\mathsf{T}^{\mathsf{Q}}_{\mu_k}\right)^m Q_{k-1}(s^*,a^*)\mathbf{1}\right)(s,a) - 2\delta_k \\
&= \left(\mathsf{T}^{\mathsf{Q}}_{\mu_k}\left(\lim_{m\to\infty}\left(\mathsf{T}^{\mathsf{Q}}_{\mu_k}\right)^m Q_{k-1}\right) - \lim_{m\to\infty}\left(\mathsf{T}^{\mathsf{Q}}_{\mu_k}\right)^m Q_{k-1}\right)(s,a) - 2\delta_k
\end{aligned}
$$

Since $\mathsf{T}^{\mathsf{Q}}_{\mu_k}$ is a continuous operator, it is possible to take the limit outside. We thus obtain,

$$
\begin{aligned}
g_k(s,a) &\geq \lim_{m\to\infty} \left(\left(\mathsf{T}^{\mathsf{Q}}_{\mu_k}\right)^{m+1} Q_{k-1} - \left(\mathsf{T}^{\mathsf{Q}}_{\mu_k}\right)^m Q_{k-1}\right)(s,a) - 2\delta_k \\
&= \lim_{m\to\infty} \left(\left(\mathsf{T}^{\mathsf{Q}}_{\mu_k}\right)^m \mathsf{T}^{\mathsf{Q}}_{\mu_k} Q_{k-1} - \left(\mathsf{T}^{\mathsf{Q}}_{\mu_k}\right)^m Q_{k-1}\right)(s,a) - 2\delta_k \\
&= \lim_{m\to\infty} \left(\mathbb{Q}^m_{\mu_k} \left(\mathsf{T}^{\mathsf{Q}}_{\mu_k} Q_{k-1} - Q_{k-1}\right)\right)(s,a) - 2\delta_k \\
&= \mathbb{Q}^*_{\mu_k} \left(\mathsf{T}^{\mathsf{Q}}_{\mu_k} Q_{k-1} - Q_{k-1}\right)(s,a) - 2\delta_k.
\end{aligned}
$$

Since $\left\|\mathsf{T}^{\mathsf{Q}}_{\mu_k} Q_{k-1} - \mathsf{T}^{\mathsf{Q}} Q_{k-1}\right\|_\infty \leq \epsilon_{k-1}$,

$$
\begin{aligned}
g_k(s,a) &\geq \mathbb{Q}^*_{\mu_k} \left(\mathsf{T}^{\mathsf{Q}} Q_{k-1} - Q_{k-1}\right)(s,a) - 2\delta_k - \epsilon_{k-1} \\
&= \left(\mathbb{Q}^*_{\mu_k} g_{k-1}\right)(s,a) - 2\delta_k - \epsilon_{k-1}.
\end{aligned}
$$

Recall the definition of $\gamma > 0$ in Lemma 4.1. Note that all entries of $\mathbb{Q}^*_{\mu_k}$ are bounded from below by $\gamma$. Hence, we have

$$
g_k(s,a) \geq (1-\gamma)l_{k-1} + \gamma u_{k-1} - 2\delta_k - \epsilon_{k-1}.
$$

Since the above inequality is true for all states $(s, a) \in \mathcal{S} \times \mathcal{A}$, we obtain,

$$l_k \geq (1 - \gamma)l_{k-1} + \gamma u_{k-1} - 2\delta_k - \epsilon_{k-1}.$$

The above inequality combined with Equation (11), yields

$$l_k \geq (1 - \gamma)l_{k-1} + \gamma J^* - 2\delta_k - \epsilon_{k-1}.$$

Upon rearranging the above inequality, we obtain the result, that is,

$$(J^* - l_k) \leq (1 - \gamma)(J^* - l_{k-1}) + 2\delta_k + \epsilon_{k-1}.$$

Subsequently,

$$\left(J^* - \min_{(s,a)} \left(\mathsf{T}^{\mathsf{Q}} Q_k - Q_k\right)(s, a)\right) \leq (1 - \gamma)\left(J^* - \min_{(s,a)} \left(\mathsf{T}^{\mathsf{Q}} Q_{k-1} - Q_{k-1}\right)(s, a)\right)$$
$$+ 2\|Q_k - Q_{\mu_k}\|_\infty + \left\|\mathsf{T}^{\mathsf{Q}} Q_{k-1} - \mathsf{T}^{\mathsf{Q}}_{\mu_k} Q_{k-1}\right\|_\infty$$

### 3.3   Proof of Proposition 4.2

We now prove Proposition 4.2. From Lemma 4.1, we have

$$(J^* - l_k) \leq (1 - \gamma)(J^* - l_{k-1}) + 2\delta_k + \epsilon_{k-1}. \tag{13}$$

To iterate over Equation (13), we need to ensure the terms are non-negative. Note that by the first part,

$$l_k - \epsilon_k \leq J^*.$$

Hence, rearranging the terms thus yields,

$$(J^* - l_k + \epsilon_k) \leq (1 - \gamma)(J^* - l_{k-1} + \epsilon_{k-1}) + \epsilon_k + 2\delta_k + \epsilon_{k-1} - (1 - \gamma)\epsilon_{k-1}.$$

This is equivalent to

$$(J^* - l_k + \epsilon_k) \leq (1 - \gamma)(J^* - l_{k-1} + \epsilon_{k-1}) + (\epsilon_k + \gamma\epsilon_{k-1}) + 2\delta_k.$$

Let

$$a_k = J^* - l_k + \epsilon_k.$$

Then, we obtain,

$$\begin{aligned}
a_k &\leq (1 - \gamma)a_{k-1} + \epsilon_k + \gamma\epsilon_{k-1} + 2\delta_k \\
&\leq (1 - \gamma)\left[(1 - \gamma)a_{k-2} + \epsilon_{k-1} + \gamma\epsilon_{k-2} + 2\delta_{k-1}\right] + \epsilon_k + \gamma\epsilon_{k-1} + 2\delta_k \\
&= (1 - \gamma)^k a_0 + \sum_{\ell=0}^{k-1}(1 - \gamma)^\ell \epsilon_{k-\ell} + \gamma \sum_{\ell=0}^{k-1}(1 - \gamma)^\ell \epsilon_{k-1-\ell} + 2\sum_{\ell=0}^{k-1}(1 - \gamma)^\ell \delta_{k-\ell}
\end{aligned}$$

Since $\epsilon_0 = 0$,

$$a_k \leq (1 - \gamma)^k a_0 + \epsilon_k + \sum_{\ell=1}^{k-1}(1 - \gamma)^{\ell-1}\epsilon_{k-\ell} + 2\sum_{\ell=0}^{k-1}(1 - \gamma)^\ell \delta_{k-\ell}$$

Substituting for $a_k$, we get

$$(J^* - l_k) \leq (1 - \gamma)^k (J^* - l_0) + \sum_{\ell=1}^{k-1}(1 - \gamma)^{\ell-1}\epsilon_{k-\ell} + 2\sum_{\ell=0}^{k-1}(1 - \gamma)^\ell \delta_{k-\ell}.$$

Since from part 1, we know that $J^* - J_{\mu_{k+1}} \leq J^* - l_k + \epsilon_k$, we have

$$J^* - J_{\mu_{k+1}} \leq (1 - \gamma)^k \left[J^* - \min_i \left(\mathsf{T}^{\mathsf{Q}} Q_0 - Q_0\right)(i)\right]$$
$$+ \sum_{\ell=1}^{k-1}(1 - \gamma)^{\ell-1}\left[\left\|\mathsf{T}^{\mathsf{Q}}_{\mu_{k+1-\ell}} Q_{k-\ell} - \mathsf{T}^{\mathsf{Q}} Q_{k-\ell}\right\|_\infty\right] + \left\|\mathsf{T}^{\mathsf{Q}}_{\mu_{k+1}} Q_k - \mathsf{T}^{\mathsf{Q}} Q_k\right\|_\infty$$
$$+ 2\sum_{\ell=0}^{k-1}(1 - \gamma)^\ell \left[\|Q_{k-\ell} - Q_{\mu_{k-\ell}}\|_\infty\right].$$

Taking expectation, we have

$$\mathbb{E}\left[J^* - J_{\mu_{k+1}}\right] \le (1-\gamma)^k \,\mathbb{E}\left[J^* - \min_i \left(\mathsf{T}^{\mathsf{Q}} Q_0 - Q_0\right)(i)\right]$$

$$+ \sum_{\ell=1}^{k-1} (1-\gamma)^{\ell-1} \,\mathbb{E}\left[\left\|\mathsf{T}^{\mathsf{Q}}_{\mu_{k+1-\ell}} Q_{k-\ell} - \mathsf{T}^{\mathsf{Q}} Q_{k-\ell}\right\|_\infty\right] + \mathbb{E}\left[\left\|\mathsf{T}^{\mathsf{Q}}_{\mu_{k+1}} Q_k - \mathsf{T}^{\mathsf{Q}} Q_k\right\|_\infty\right]$$

$$+ 2 \sum_{\ell=0}^{k-1} (1-\gamma)^\ell \,\mathbb{E}\left[\left\|Q_{k-\ell} - Q_{\mu_{k-\ell}}\right\|_\infty\right].$$

## 3.4  TD Learning for $Q$ function

One RL algorithm to estimate the state-action relative value function with linear function approxima-tion, associated with a policy $\mu$ is TD($\lambda$). Finite-time sample complexity of TD($\lambda$) has been recently studied in [ZZM21]. Invoking their results, restating it in terms of the state-action relative value function rather than the state relative value function, we characterize the sample complexity required to achieve certain accuracy in the approximation.

We estimate $Q_\mu(s,a)$ linearly by $\phi(s,a)^\top \theta_\mu^*$, for some $\theta_\mu^* \in \mathbb{R}^d$, where $\phi(s,a) = [\phi_1(s,a), \cdots, \phi_d(s,a)]^T \in \mathbb{R}^d$ is the feature vector associated with $(s,a) \in \mathcal{S} \times \mathcal{A}$. Here, $\theta_\mu^*$ is a fixed point of $\Phi\theta = \Pi_{D,W_\phi} \mathsf{T}^{\mathsf{Q},\lambda}_\mu \Phi\theta$, where $\Phi$ is an $|\mathcal{S}| \times d$ matrix whose $k$-th column is $\phi_k$, $\mathsf{T}^{\mathsf{Q},\lambda}_\mu = (1-\lambda)\sum_{m=0}^\infty \lambda^m \left(\widehat{\mathsf{T}}^{\mathsf{Q}}_\mu\right)^{m+1}$ where $\widehat{\mathsf{T}}^{\mathsf{Q}}_\mu = \mathsf{T}^{\mathsf{Q}}_\mu - J_\mu \mathbf{1}$, $D$ is the diagonal matrix with diagonal elements given by the stationary distribution of the policy $\mu$ on $\mathcal{S} \times \mathcal{A}$, and $\Pi_{D,W_\phi} = \Phi(\Phi^\top D\Phi)^{-1}\Phi^\top D$ is the projection matrix onto $W_\Phi = \{\Phi\theta | \theta \in \mathbb{R}^d\}$ with respect to the norm $\|\cdot\|_D$. Note that this fixed point equation may have multiple solutions. In particular, if $\Phi\theta_e = \mathbf{1}$, then $\theta_\mu^* + p\theta_e$ is also a solution for any $p \in \mathbb{R}$. Hence, we focus on the set of equivalent classes $E$ where we say $\theta_1 \sim \theta_2$ if $\Phi(\theta_1 - \theta_2) = \mathbf{1}$. The value of $\mathbb{E}[\|\Pi_{2,E}(\theta - \theta_\mu^*)\|_2^2]$ characterizes the accuracy of our approximation, where $\Pi_{2,E}$ is the projection onto $E$ with respect to $\|\cdot\|_2$. Suppose that $\{\phi_1, \phi_2, \cdots, \phi_d\}$ are linearly independent and that $\max_{(s,a)\in\mathcal{S}\times\mathcal{A}} \|\phi(s,a)\|_2 \le 1$.

Upon making a new observation $(s_{t+1}, a_{t+1})$ for $t \ge 0$, the average reward $TD(\lambda)$ uses the following update equations:

$$\begin{aligned}
&\text{Eligibility trace: } z_t = \lambda z_{t-1} + \phi(s_t, a_t)\\
&\text{TD error: } d_t = r(s_t, a_t) - J_t + \phi(s_{t+1}, a_{t+1})^\top \theta_t\\
&\qquad\qquad\qquad - \phi(s_t, a_t)^\top \theta_t \\
&\text{average-reward update: } J_{t+1} = J_t + c_\alpha \beta_t(r(s_t, a_t) - J_t),\\
&\text{parameter update: } \theta_{t+1} = \theta_t + \beta_t \delta_t(\theta) z_t,
\end{aligned} \qquad (14)$$

where $\beta_t$ is the scalar step size, $c_\alpha > 0$ is a constant, and $z_{-1}$ is initialized to be zero. Following the same argument as in [ZZM21], we get the following theorem which characterizes the number of samples required to get a certain accuracy in the approximation.

**Theorem 3.1.** *Let $\beta_t = \frac{c_1}{t+c_2}$, and suppose that the positive constants $c_1$, $c_2$, and $c_\alpha$ are properly chosen. Then, the number of samples required to ensure an approximation accuracy of $\mathbb{E}[\|\Pi_{2,E}(\theta - \theta_\mu^*)\|_2^2] \le \delta$, is given by*

$$\tau = O\left(\frac{K\log(\frac{1}{\Delta})\|\theta_\mu^*\|_2^2}{\Delta^4(1-\lambda)^4\delta^2}\right),$$

*where $K$ is the minimum mixing time constant across all probability transition matrices induced by the policies,*

$$\Delta = \min_{\|\theta\|_2=1, \theta\in E} \theta^T \Phi^T D(I - \mathbb{Q}^\lambda)\Phi\theta > 0,$$

*and $\mathbb{Q}^\lambda = (1-\lambda)\sum_{m=0}^\infty \lambda^m \mathbb{Q}^{m+1}$.*

*Proof.* The proof is a simple adaptation of the proof in [ZZM21] for the state relative value function. □

Let $Q_k = \Phi\theta_k$, where $\theta_k$ is the output of TD($\lambda$) algorithm with $T$ samples. Then, we have

$$\|Q_k - Q_{\mu_k}\|_\infty \leq \|\Phi(\theta_k - \theta^*_{\mu_k})\|_\infty + \|\Phi\theta^*_{\mu_k} - Q_{\mu_k}\|_\infty,$$

where the second term above is the error associated with the choice of $\Phi$.

Since $\max_{(s,a)\in\mathcal{S}\times\mathcal{A}} \|\phi(s,a)\|_2 \leq 1$, we get

$$\delta_k = d\|\theta_k - \theta^*_{\mu_k}\|_\infty + \|\Phi\theta^*_{\mu_k} - Q_{\mu_k}\|_\infty.$$

Using Theorem 3.1, we get a bound on the value of $\|\theta_k - \theta^*_{\mu_k}\|_\infty$. Note that if $\Phi\theta_e = \mathbf{1}$ for some $\theta_e \in \mathbb{R}^d$, then we can pick $\theta_{k+1}$ and $\theta^*_{\mu_{k+1}}$ so that $\phi(s^*, a^*)^T \theta_{k+1} = \phi(s^*, a^*)^T \theta^*_{\mu_{k+1}} = 0$. Otherwise, there is a unique choice for $\theta^*_{\mu_{k+1}}$, and we have $E = \mathbb{R}^d$.

The expected value of learning error can thus be expressed as:

$$\mathbb{E}[\delta_k] = \mathbb{E}[\delta_{\mathsf{TD},k}] + \mathbb{E}[\delta_{0,k}]$$

where $\delta_{\mathsf{TD},k} = d\|\theta_{k+1} - \theta^*_{\mu_{k+1}}\|_\infty$ represents the TD learning error of the parameter vector $\theta^*_{\mu_{k+1}}$ and $\delta_{0,k} = \|\Phi\theta^*_{\mu_{k+1}} - Q_{\mu_{k+1}}\|_\infty$ represents the function approximation error associated with the span of the feature vector matrix $\Phi$.

$\delta_{\mathsf{TD},k}$ has a direct dependence on the number of samples utilized for TD learning and the mixing time of the corresponding Markov Chain induced by the policy $\mu_k$. More precisely, from Theorem 3.1,

$$\mathbb{E}[\delta_{\mathsf{TD},k}] = O\left(\sqrt{\frac{K\log(\frac{1}{\Delta})\|\theta^*_{\mu_k}\|_2^2}{\Delta^4(1-\lambda)^4\tau}}\right),$$

Hence as long as the mixing constant $\Delta$ is uniformly greater than zero, and the feature vectors $\theta^*_{\mu_k}$ are such that they are uniformly upper bounded in $k$, it is true that

$$\mathbb{E}[\delta_{\mathsf{TD},k}] = \frac{C}{\sqrt{\tau}},$$

for some constant $C > 0$.

Next, we prove the corollaries for specific policy improvement algorithms. Since the policy improvement part does not depend on whether the problem is a discounted-reward problem or an average reward problem, we can borrow the results from [CM22] to identify $\epsilon_k$ in each of the corollaries.

### 3.5 Proof of Corollary 4.3

Recall the greedy update rule. Let $a^* = \arg\max_{a'} Q_k(s, a')$. Given a parameter $\beta > 0$, at any time instant $k$, the greedy policy update $\mu_{k+1}$ is given by:

$$\mu_{k+1}(a|s) = \begin{cases} \frac{1}{\beta|\mathcal{A}|}, & \text{if } a \neq a^* \\ \frac{1}{\beta|\mathcal{A}|} + 1 - \frac{1}{\beta}, & \text{if } a = a^* \end{cases} \tag{15}$$

Let $\eta_k = \max_{\substack{s'\in\mathcal{S}\\a'\in\mathcal{A}}} |\mathbb{Q}_k(s', a')|$. The policy improvement approximation can be shown to be the following:

$$\begin{aligned} \epsilon_k &= \left(\mathsf{T}^Q Q_k - \mathsf{T}^Q_{\mu_{k+1}} Q_k\right)(s, a) \\ &\leq \sum_{s'\in\mathcal{S}} \mathbb{P}(s'|s, a)\frac{2}{\beta}\max_{a'\in\mathcal{A}}|Q_k(s', a')| \\ &\leq \frac{2\eta_k}{\beta}. \end{aligned}$$

Recall the error due to TD Learning

$$\mathbb{E}[\delta_k] = \mathbb{E}[\delta_{\text{TD},k}] + \mathbb{E}[\delta_{0,k}].$$

Let $\overline{\delta}_0 = \max_t \delta_{0,t}$. Then we obtain the following:

$$\mathbb{E}[\delta_k] = \overline{\delta}_0 + \frac{C}{\sqrt{\tau}}.$$

Substituting for expressions in Proposition 4.2, we obtain,

$$\mathbb{E}\left[J^* - J_{\mu_{T+1}}\right] \le (1-\gamma)^T \mathbb{E}\left[J^* - \min_i \left(\mathsf{T}^Q Q_0 - Q_0\right)(i)\right]$$
$$+ 2\sum_{\ell=0}^{T-1}(1-\gamma)^\ell \left(\overline{\delta}_0 + \frac{C}{\sqrt{\tau}}\right) + \sum_{\ell=1}^{T-1}(1-\gamma)^{\ell-1}\frac{2\eta_{T-\ell}}{\beta} + \frac{2\eta_T}{\beta}.$$

Let $c_0 = \mathbb{E}\left[J^* - \min_i \left(\mathsf{T}^Q Q_0 - Q_0\right)(i)\right]$ be the error associated with the initial condition. Let $\overline{\eta} = \max_t \eta_t$ ($\overline{\eta}$ can be the uniform upper bound of the estimates $Q_k$ of relative state action value function $Q_{\mu_k}$ over $k$.) Then we obtain the result in the corollary,

$$\mathbb{E}\left[J^* - J_{\mu_{T+1}}\right] \le (1-\gamma)^T c_0 + \frac{2}{\gamma}\left(\overline{\delta}_0 + \frac{C}{\sqrt{\tau}}\right) + \left(\frac{1+\gamma}{\gamma}\right)\frac{2\overline{\eta}}{\beta}. \tag{16}$$

### 3.6 Proof of Corollary 4.4

Recall the softmax policy update. Given a parameter $\beta > 0$, at any time instant $k$, the Softmax update policy $\mu_{k+1}$ is:

$$\mu_{k+1}(a|s) = \frac{\exp\left(\beta Q_k(s,a)\right)}{\sum_{a' \in \mathcal{A}} \exp\left(\beta Q_k(s,a')\right)}. \tag{17}$$

The policy improvement approximation for this update rule turns out to be time independent and is given by:

$$\epsilon_k = \left(\mathsf{T}^Q Q_k - \mathsf{T}^Q_{\mu_{k+1}} Q_k\right)(s,a) \le \frac{\log|\mathcal{A}|}{\beta}.$$

Given $\overline{\delta}_0 = \max_t \delta_{0,t}$ the following is true,

$$\mathbb{E}[\delta_k] = \overline{\delta}_0 + \frac{C}{\sqrt{\tau}}.$$

Substituting for expressions in Proposition 4.2, we obtain,

$$\mathbb{E}\left[J^* - J_{\mu_{T+1}}\right] \le (1-\gamma)^T \mathbb{E}\left[J^* - \min_i \left(\mathsf{T}^Q Q_0 - Q_0\right)(i)\right]$$
$$+ 2\sum_{\ell=0}^{T-1}(1-\gamma)^\ell \left(\overline{\delta}_0 + \frac{C}{\sqrt{\tau}}\right) + \sum_{\ell=1}^{T-1}(1-\gamma)^{\ell-1}\frac{\log|\mathcal{A}|}{\beta} + \frac{\log|\mathcal{A}|}{\beta}.$$

Let $c_0 = \mathbb{E}\left[J^* - \min_i \left(\mathsf{T}^Q Q_0 - Q_0\right)(i)\right]$ be the error associated with the initial condition. Then we obtain the result in the corollary,

$$\mathbb{E}\left[J^* - J_{\mu_{T+1}}\right] \le (1-\gamma)^T c_0 + \frac{2}{\gamma}\left(\overline{\delta}_0 + \frac{C}{\sqrt{\tau}}\right) + \left(\frac{1+\gamma}{\gamma}\right)\frac{\log(|\mathcal{A}|)}{\beta}. \tag{18}$$

### 3.7 Proof of Corollary 4.5

Recall the mirror descent update. Given $\beta > 0$ the mirror descent update is given by:

$$\mu_{k+1}(a|s) = \frac{\mu_k(a|s)e^{\beta Q_k(s,a)}}{\sum_{a' \in \mathcal{A}} \mu_k(a'|s)e^{\beta Q_k(s,a')}} \tag{19}$$

The policy improvement error for this update rule and is given by:

$$\epsilon_k = \left(\mathsf{T}^Q Q_k - \mathsf{T}^Q_{\mu_{k+1}} Q_k\right)(s,a)$$

$$\leq \frac{1}{\beta} \log \frac{1}{\min_{s \in \mathcal{S}} \mu_{k+1}\left(a^*(s)|s\right)}.$$

where $a^*(s)$ is the optimal action at state $s$. Let $\omega_{k+1} = \min_{s \in \mathcal{S}} \mu_{k+1}\left(a^*(s)|s\right)$ Given that the TD learning error is of the form

$$\mathbb{E}\left[\delta_k\right] = \bar{\delta}_0 + \frac{C}{\sqrt{\tau}},$$

Substituting for expressions in Proposition 4.2, we obtain,

$$\mathbb{E}\left[J^* - J_{\mu_{T+1}}\right] \leq (1-\gamma)^T \mathbb{E}\left[J^* - \min_i \left(\mathsf{T}^Q Q_0 - Q_0\right)(i)\right]$$

$$+ 2\sum_{\ell=0}^{T-1} (1-\gamma)^\ell \left(\bar{\delta}_0 + \frac{C}{\sqrt{\tau}}\right) + \sum_{\ell=1}^{T-1} (1-\gamma)^{\ell-1} \frac{2}{\beta} \log \frac{1}{\omega_{T-\ell}} + \frac{2}{\beta} \log \frac{1}{\omega_T}.$$

Let $c_0 = \mathbb{E}\left[J^* - \min_i \left(\mathsf{T}^Q Q_0 - Q_0\right)(i)\right]$ be the error associated with the initial condition. Let $\underline{\omega} = \min_t \omega_t$ Then we obtain the result in the corollary,

$$\mathbb{E}\left[J^* - J_{\mu_{T+1}}\right] \leq (1-\gamma)^T c_0 + \frac{2}{\gamma}\left(\bar{\delta}_0 + \frac{C}{\sqrt{\tau}}\right) + \left(\frac{1+\gamma}{\gamma\beta}\right)\log\left(\frac{1}{\underline{\omega}}\right). \tag{20}$$

## 3.8 Regret Analysis

Recall the pseudo regret defined as follows:

$$R_{\text{PS}}(K) = \sum_{t=1}^{K} \left(J^* - J_{\mu_t}\right), \tag{21}$$

where $\mu_t$ us the policy utilized at time $t$ and obtained through mirror descent, and $K$ is the time horizon. Since $K = \tau T$, we have

$$R_{\text{PS}}(K) = \tau \sum_{t=1}^{T} \left(J^* - J_{\mu_t}\right). \tag{22}$$

Substituting for $J^* - J_{\mu_t}$, it is true that

$$R_{\text{PS}}(K) = \tau \left(\sum_{t=1}^{T} (1-\gamma)^T c_0 + \frac{1+\gamma}{\gamma\beta} \log\left(\frac{1}{\underline{\omega}}\right) + \frac{2}{\gamma}\left(\bar{\delta}_0 + \frac{c_3}{\sqrt{\tau}}\right)\right). \tag{23}$$

Let $\beta = \sqrt{\tau}$. The above regret can be simplified as:

$$R_{\text{PS}}(K) = \frac{\tau c_0}{\gamma} + \frac{K}{\gamma\sqrt{\tau}}\left((1+\gamma)\log\left(\frac{1}{\underline{\omega}}\right) + 2c_3\right) + \frac{2K\bar{\delta}_0}{\gamma} \tag{24}$$

Let $c_5 = (1+\gamma)\log(1/\underline{\omega}) + 2c_3$. We then have

$$R_{\text{PS}}(K) = \frac{\tau c_0}{\gamma} + \frac{K c_5}{\gamma\sqrt{\tau}} + \frac{2K\bar{\delta}_0}{\gamma}. \tag{25}$$

Optimizing for regret involves equating $\frac{\tau c_0}{\gamma}$ and $\frac{K c_5}{\gamma\sqrt{\tau}}$. This yields $\tau = \left(\frac{K c_5}{c_0}\right)^{2/3}$. Further substituting for $\tau$ yields

$$R_{\text{PS}}(K) = \left(\frac{K c_3}{c_0}\right)^{2/3} \cdot \frac{c_0}{\gamma} + \frac{K c_5 c_0^{1/3}}{\gamma (K c_5)^{1/3}} + \frac{2K\bar{\delta}_0}{\gamma} \tag{26}$$

$$= \frac{(K c_5)^{2/3} \cdot c_0^{1/3}}{\gamma} + \frac{(K c_5)^{2/3} \cdot c_0^{1/3}}{\gamma} + \frac{2K\bar{\delta}_0}{\gamma} \tag{27}$$

$$= \frac{2 c_5^{2/3} \cdot c_0^{1/3} \cdot K^{2/3}}{\gamma} + \frac{2K\bar{\delta}_0}{\gamma}. \tag{28}$$

Let $c_6 = 2c_5^{2/3}c_0^{1/3}$. We have

$$R_{\text{PS}}(K) = \frac{K^{2/3}c_6}{\gamma} + \frac{2K\overline{\delta_0}}{\gamma}. \tag{29}$$