# OpenReview forum: "Performance Bounds for Policy-Based Average Reward Reinforcement Learning Algorithms"
_NeurIPS.cc/2023/Conference — NeurIPS 2023 poster_

### Official Review · Reviewer_M1AZ · 2023-06-28

**Soundness:** 3 good
**Presentation:** 3 good
**Contribution:** 3 good
**Rating:** 7
**Confidence:** 5

**Summary:**

This paper derives some performance bounds on some approximate policy iteration type algorithms and later considers the application in the context of reinforcement learning. The bounds derived appear correct and meaningful but I have several comments which I will list below.

**Strengths:**

This paper derives some performance bounds on some approximate policy iteration type algorithms and later considers the application in the context of reinforcement learning. The bounds derived appear correct and meaningful but I have several comments which I will list below.

**Weaknesses:**

One of the weaknesses is a lack of discussion on previous bounds that are available in the context of average reward problems.
A central class of algorithms in the RL literature based on the policy iteration method have been the actor-critic algorithms.
 I would have preferred a thorough comparison and discussion on the significance of the bounds that the authors derive and how different are these from the bounds obtained for the actor-critic schemes.

A second weakness is of the lack of any experimentation. That would have certainly added value to the paper even though the primary focus of the authors has been to derive some bounds for the algorithms.

A third weakness is the lack of a comprehensive analysis of asymptotic convergence for these algorithms.

**Questions:**

1. In the discussion following (1), the authors try to argue out a bound for average reward problems after stating a bound for the discounted reward problems. Do we really have to do it this way to argue out the problems that might result. Why do we even need to use the bounds for discounted MDPs in the average reward setting?

2. For the equation above (2), the authors would need to assume certain conditions on the underlying Markov chain in order for the limit to exist.

3. Para above Sec 2.2 has incorrect statements. It suggests that algorithms based on approximate DP for average reward MDPs do not exist. The authors should look up references such as [BSGL09] and [KT99] cited their paper. There are several other papers that deal with average reward MDPs and use function approximation in particular.

4. In Algorithm 1, the authors should specify what \epsilon and \delta are. Also, in step 2, how do you decide on J_{k+1}? Is there a systematic way of it's selection.
5. In Theorem 3.3, is \gamma the same as in Lemma 3.2? Mention also whether \gamma >0.

6. Second line after Corollary 3.4: Substituting \gamma = (1-\alpha)^2 makes the denominator same in (4)-(5) but numerators are quite different. So, not clear what the authors mean by saying \gamma plays the role of (1-\alpha)^2.

7. In Lemma 4.1, what is T? Is this the time of episode termination. Also, is T deterministic or random? How is T chosen and what is the effect of T on the performance since it is an important parameter?

8. Is the \gamma in Lemma 4.1 the same as \gamma in Lemma 3.2? I guess that is not true because this \gamma is based on the state-action Markov chain unlike the state-valued chain previously. Why should the invariant distribution of the state-action Markov chain with transition probability matrix Q_\mu exist?

9. How does the inequality (6) hold? What is Q_k on the LHS and how did it vanish on the RHS?

10. In Corollaries 4.3-4.5, what is the parameter \tau and is it related to T? One would have expected the inequalities in these results to go to 0 as T goes to infinity, but that is not happening. Why does this problem arise and so how meaningful are the presented bounds in these corollaries?

11. In (15), the RHS is multiplied by \tau which was not the case earlier (see line below (14)). This makes the role of \tau confusing?

*********
The following comments are post the rebuttal from the authors.

1. I am again looking at the paper. For one, things are not well presented at all. For instance, Algorithm 3 does not involve a theta_k parameter but in (6), the terms \theta_k, \theta^*_{\mu_k} appear on the RHS saying that Algorithm 3 gives this. The authors begin by talking of approximate policy iteration and then say in the case of RL, the terms \theta_k etc appear and there is no algorithm given for the update of the theta_k.

2. The authors say that they are able to prove convergence to global optima even with function approximation but it's not clear how? In Remark 4.6, they mention that there is an error due to function approximation which is also the case with other algorithms in the literature that use function approximation.

3. Also, the authors say they show asymptotic convergence. Is the convergence shown in the almost sure sense? If so, where is that result? Also, somewhere they said that [ZZM21] showed the bound in (7) assuming that \theta^*_{\mu_k} are uniformly bounded. It is not stated if the authors also assume this bound. Also, what about a bound on the \theta_k sequence itself? Do you not require to prove stability of your stochastic iterate before claiming asymptotic convergence?

4. Assumption 3.1a clearly is too strong because it is only for the case of deterministic policies. Following 3.1a, the authors say that they consider instead a modified policy where with some positive probability, they select an action uniformly at random and with the remaining probability, they select actions as per a deterministic policy. Clearly, this is not a deterministic policy. Then the assumption makes no sense. I find this whole thing a rather strange argument.

5. Assumption 3.1b also requires a special structure on the Markov chain which may not always be the case - that the diagonal elements in the probability matrix have non-zero entries under every policy. This is also the reason why experiments are important. The authors can easily demonstrate in a future version of their work whether such unrealistic assumptions are needed even for empirical studies.
*****

The authors have been able to convince me that theirs is a good contribution. I am therefore revising my score.


**Limitations:**

I have already listed my comments above.

---

> ### Author Rebuttal · Authors · 2023-08-07
>
> We thank the reviewer for their review and comments. Please find our response to the weaknesses:
>
> 1. Prior work such as [KT99] and [BSGL09] are mostly based on the analysis of policy gradient (PG). Since PG typically only converges to a local minimum, they only provide local convergence results. Moreover, even the local convergence results are only asymptotic in nature and do not have finite-time bounds. To the best of our knowledge, the only works close to our paper are [AYBB+19] and [LYAYS21], where they analyze a specific form of actor critic algorithm (mirror descent based). Section 4.2 provides a comparison of the bounds in our paper with the performance bounds in [AYBB+19] and [LYAYS21]. Specifically, we point out that in the case of the most widely used algorithm for policy evaluation, i.e., TD-learning, our approach via approximate PI improves the regret bounds from $O(T^{3/4})$ to $O(T^{2/3}).$
>
> 2. Since the aim of the paper was to provide finite time performance bounds for a general class of actor-critic algorithms, we did not think experiments would have contributed towards this goal.
>
> 3. Section 4 deals with performance bounds for RL algorithms. There by letting the number of iterations go to infinity, one immediately gets asymptotic performance bounds. As is typically the case in RL algorithms, the asymptotic error can only be made to go to zero if the feature vectors are rich enough to make the function approximation error go to zero. For example, in Corollary 4.3, Equation (9), the asymptotic error can be seen to be small if the function approximation error $\bar{\delta_0}=0$ and $\beta\to\infty$. We provide finite-time bounds because they are more realistic (and the asymptotic error can be recovered from them by letting $T\to\infty$) and they provide insight into the tradeoffs involved in choosing various parameters. For example, making $\beta$ large may increase $C$ as can be seen from [ZZM21]. However, under our assumptions, $C$ will be bounded and so, one cannot see its impact by looking at asymptotic expressions alone.
>
> Response to questions:
>
> 1. The point of the discussion was to illustrate the difficulty in getting bounds for the average reward case. In particular, as $\alpha\to1,$ it is well known that one can recover average-reward results from discounted reward results. The discussion points out that this relationship fails to provide a meaningful result in the case of avg reward approximate PI.
>
> 2. The ergodicity assumption (Assumption 3.1) in the paper ensures that this limit exists.
>
> 3. We agree with the reviewer, but note some important differences between prior work and ours. As mentioned before, the references [KT99] and [BSGL09] only provide convergence to a local optimum and the results are asymptotic in nature, i.e., no finite-time performance bounds. And the algorithms studied there are only gradient-based algorithms. On the other hand, our goal is to provide a unified framework for studying many algorithms, and to obtain finite-time performance bounds.
>
> 4. $\epsilon$ and $\delta$ are the upper bounds on the policy improvement and policy evaluation errors respectively (as indicated in Algorithm 1). $J_{k+1}$ is determined based on the policy learning algorithm utilized. We note that approximate PI is a framework to study RL algorithms and is not an RL algorithm by itself. One has to then relate them to RL algorithms so that we can use bounds on policy-based RL methods by showing that they are instantiations of approximate PI.
>
> 5. The two $\gamma$'s are identical. Lines 191 and 209 both indicate that $\gamma>0$.
>
> 6. Even though the constants in the numerator of Equation 4 and 5 are different, the growth rate of the RHS in $\epsilon$ and $\delta$ is the same. In particular, both grow linearly with policy improvement and policy approximation errors.
>
> 7. $T$ represents the number of iterations of Algorithm 3. The results in Lemma 4.1,  Proposition 4.2 and Corollaries 4.3-4.5 hold for any $T$. $T$ is any deterministic integer. The lemma, the proposition and the corollaries provide finite-time performance guarantees as a function of $T.$ In Section 4.2, we further provide regret bounds in terms of the total number of samples used, which is $T$ multiplied by the number of time steps in each policy evaluation step.
>
> 8. $\gamma$ in Lemma 4.1 is defined within the same lemma and it isn't the same as the one in Lemma 3.2. Since we are considering randomized policies and under the ergodicity assumption 3.1, the invariant distribution across state-action pairs do exist. We refer the reviewer to the Appendix on Discussion on Assumption for more details.
>
> 9. $Q_k$ is the approximation to the state-action value function $Q_{\mu_k}$ (defined in Algorithm 3). Since $Q_k$ is learnt through TD learning with linear function approximation, $Q_k=\Phi\theta_k$. Inequality 6 decomposes the evaluation error into function approximation error and estimation error. For more details refer lines 247-256.
>
> 10. $\tau$ is the length of the sample trajectory utilized for evaluating a policy. It is defined in the first sentence of Section 4.1, $\tau$ and $T$ are independent of another. For every iteration of Algorithm 3, the learning step 2 involves generating a trajectory of length $\tau.$ The RHS of the corollaries will not go to 0 as $T$ tends to infinity, since the approximation errors in policy evaluation may not go to 0. Similar to corollary 3.4, the asymptotic bounds in corollaries 4.3-4.5 do not go to zero as long as the approximation errors are finite. This is standard in RL theory. However, we note that the errors will go to zero if the feature space is rich enough to represent the value functions and if we choose $\beta$ to be sufficiently large.
>
> 11. For the purpose of regret analysis it is necessary to consider the total time units, which here happens to be $T\tau$ (since for every iteration of Algorithm 3, a sample trajectory of length $\tau$ is generated).

---

> > ### Comment · Reviewer_M1AZ · 2023-08-15
> > **Further comments from M1AZ**
> >
> > I have seen the rebuttal from the authors. I am not convinced about their contribution. Just deriving finite time bounds for the average reward actor-critic scheme is not good enough particularly because this is not the first paper in this space. For instance, in the following paper ``N.Saxena, S.Khastagir, S.Kolathaya and S.Bhatnagar, Off-Policy Average Reward Actor-Critic with Deterministic Policy Search", ICML, 2023, in addition to finite time bounds, the authors show a detailed asymptotic convergence analysis and also show the results of several experiments.
> > Because of the shortcoming in the contribution, I shall retain my score.

---

> > > ### Author Response · Authors · 2023-08-15
> > >
> > > Thanks for pointing out the reference. We would like to note that the reference only provides local convergence. They show that the sample complexity for getting the gradient close to $\epsilon$ is $1/\epsilon^{2.5}$. The study of local convergence of policy gradient algorithms is very old and such papers have been cited in our paper. The paper mentioned by the reviewer tweaks such algorithms to get finite-time bounds, but we emphasize that these supposed $\textbf{finite-time bounds are about getting the gradient close to zero}$ and $\textbf{not about optimality guarantees as in our paper}$. So our papers are not comparable, our results are much stronger.
> > >
> > > In the first part of our paper, approximate PI is a framework for studying policy-based algorithms. This is just a foundation for studying policy-based RL algorithms. In particular, we show that we can provide optimality guarantees for a large class of algorithms. For example, we are able to analyze mirror descent-based algorithms, softmax and $\beta$ greedy update and provide optimality guarantees, whereas the paper cited by the reviewer only considers policy gradient, which in general enjoys no global optimality guarantees whatsoever even when the value function is estimated exactly.
> > >
> > > We again emphasize that our results are much stronger than in the paper mentioned by the reviewer: We present regret bounds (not just error at one time step). For example, we  get a regret of $K^{2/3}$ for the mirror descent algorithm with TD learning, which is much stronger than even providing closeness to optimality at a particular time step. If one is just interested in just how to close to optimality we are after K steps, our result states that the gap is $K^{-1/3}.$ Clearly the above automatically yields an asymptotic result. Letting $K\to\infty$ shows that the asymptotic error is zero up to function approximation errors.

---

> > > > ### Comment · Reviewer_M1AZ · 2023-08-16
> > > > **response 2 from M1AZ**
> > > >
> > > > 1. I am again looking at the paper. For one, things are not well presented at all. For instance, Algorithm 3 does not involve a theta_k parameter but in (6), the terms \theta_k, \theta^*_{\mu_k} appear on the RHS saying that Algorithm 3 gives this. The authors begin by talking of approximate policy iteration and then say in the case of RL, the terms \theta_k etc appear and there is no algorithm given for the update of the theta_k.
> > > >
> > > > 2. The authors say that they are able to prove convergence to global optima even with function approximation but it's not clear how? In Remark 4.6, they mention that there is an error due to function approximation which is also the case with other algorithms in the literature that use function approximation.
> > > >
> > > > 3. Also, the authors say they show asymptotic convergence. Is the convergence shown in the almost sure sense? If so, where is that result? Also, somewhere they said that [ZZM21] showed the bound in (7) assuming that \theta^*_{\mu_k} are uniformly bounded. It is not stated if the authors also assume this bound. Also, what about a bound on the \theta_k sequence itself? Do you not require to prove stability of your stochastic iterate before claiming asymptotic convergence?
> > > >
> > > > 4. Assumption 3.1a clearly is too strong because it is only for the case of deterministic policies. Following 3.1a, the authors say that they consider instead a modified policy where with some positive probability, they select an action uniformly at random and with the remaining probability, they select actions as per a deterministic policy. Clearly, this is not a deterministic policy. Then the assumption makes no sense. I find this whole thing a rather strange argument.
> > > >
> > > > 5. Assumption 3.1b also requires a special structure on the Markov chain which may not always be the case - that the diagonal elements in the probability matrix have non-zero entries under every policy. This is also the reason why experiments are important. The authors can easily demonstrate in a future version of their work whether such unrealistic assumptions are needed even for empirical studies.

---

> > > > > ### Author Response · Authors · 2023-08-16
> > > > >
> > > > > We respectfully, but strongly, disagree with the reviewer's comments. We find it slightly disturbing that the reviewer is only now reading the paper carefully and asking for clarifications while incorrectly comparing our paper with the Saxena et al paper in their earlier comments: There are two major styles of analysis: (i) using stochastic approximation theory to study policy gradient algorithms and (ii) using dynamic programming, optimization and CS theory perspective (the book by Agrawal, Kakade, Jiang and Sun, https://rltheorybook.github.io/ is a good example of this perspective). The reviewer is perhaps an expert in (i) but does not appear to know the literature in (ii). So we will provide some background first.
> > > > >
> > > > > Vanilla policy gradient does not converge in general and in fact is known to converge only under very restrictive conditions (Bhandari-Russo, https://arxiv.org/pdf/1906.01786.pdf) even when the Q function can be evaluated exactly. On the other hand, variants of policy gradient which are rooted in mirror descent are known to converge globally (there are many papers that show this, please refer to Agrawal et al, Theory of policy-gradient algorithms https://www.jmlr.org/papers/volume22/19-736/19-736.pdf). Interestingly this widely cited paper is not even referred to in the Saxena et al paper, which indicates that those authors also are only familiar with the stochastic approximation approach. We note that we respect and acknowledge both approaches in our paper. In the Agrawal et al style of analysis, the goal is to bound the difference between the expected performance of two policies, and obtain error bounds and regret bounds in the expected sense. This is different from the stochastic approximation approach which aims for a.s. convergence but typically only to show that the gradient vanishes. Nevertheless, we will add a discussion in our paper with comparison to the Saxena et al paper. It is also useful to note that the Saxena et al paper appeared in iCML July 2023, so it is a little strange to expect us to know that paper.
> > > > >
> > > > > Now with respect to the comments:
> > > > >
> > > > > 1. Algorithm 3 is a generic policy based algorithm, the specific details of how each step is implemented varies depending on the specific algorithms used, which are in Section 4; the reviewer does not seem to have read the sentences around equation (6). We do not state the update equation for $\theta_k$ because it is standard TD learning and we have cited a paper as a reference [ZZM21].
> > > > >
> > > > > 2. We do not say that we prove convergence to a global optimum even with function approximation error. We stated that we prove optimality up to a function approximation error (please see the last sentence of our previous response). Again, we refer the reviewer to the Agrawal et al paper for this style of analysis.
> > > > >
> > > > > 3. No, as mentioned above, the convergence is stated in terms of the difference between the expected performance of the optimal policy and the policy obtained by algorithms which fit into the framework of policy-based methods. This is standard in the second style of analysis mentioned earlier. For our analysis, we only need a bound on $E(\|\|\theta_k-\theta^*\||^2)$ and we have to assume that $\theta^*$ is bounded across policies. One can easily put a bound on $\|\|\theta^*\|\|$ when performing TD-learning and the effect of such a bound will be reflected in the function approximation error. Parameter bounds or regularization is commonly used in almost all estimation techniques, so this is not a very restrictive assumption. Stability of stochastic iterates are shown by bounding $E(\|\|\theta_k-\theta^*\||^2)$, which is all we need. Such bounds can be obtained using Bhandari, Russo, Singal (http://proceedings.mlr.press/v75/bhandari18a/bhandari18a.pdf) and Srikant-Ying (http://proceedings.mlr.press/v99/srikant19a/srikant19a.pdf); this is the style of analysis carried out in [ZZM21] and we have cited and used their results in our paper.
> > > > >
> > > > > Responses to 4. and 5. follow below.

---

> > > > > > ### Author Response · Authors · 2023-08-16
> > > > > > **Continuation of previous comment**
> > > > > >
> > > > > > 4. Regarding the ergodicity assumption, again it appears as though the reviewer has not made an attempt to understand the assumption .We assume that in the original MDP with a small probability $\epsilon$, we take all actions with equal probability for example, and with probability $1-\epsilon$ we take some action to optimize our performance objective. Thus, we define a new MDP, which only considers the probability $1-\epsilon$ event in the original MDP, and the $\epsilon$ part is absorbed in the probability transition matrix. Now our Assumption 3.1(a) is on deterministic policies for the new MDP but this implies properties that hold for all policies including randomized policies considered in the paper. This fact is exploited throughout the paper for the randomized policies considered here. We could have simply said we assume ergodicity, our statements and assumptions state how this can be achieved at the cost of a small loss in performance, which is generally not done in other papers.
> > > > > >
> > > > > > 5. Again the reviewer does not seem to have read the paper when they comment on the diagonal entries. As we have stated in the paper and has been known since 1971, there is a transformation called the Schweitzer transformation which converts an MDP into an MDP which satisfies the assumption without loss of generality. We have provided the citation in the paragraph immediately following Assumption 3 and discussed implementation considerations in the supplementary material.

---

> > > > > > > ### Comment · Reviewer_M1AZ · 2023-08-17
> > > > > > > **response from M1AZ**
> > > > > > >
> > > > > > > Assumption 3a still appears strong to me despite the assertion otherwise by the authors. The authors provide a finite time error bound on the expected error which they argue will go to zero as the number of iterates go to infinity provided the true value function lies in the subspace of the features. This is not the stability requirement that I am referring to. I am talking of stability on sample paths. If the authors don't have a response to it, they should say it directly instead of classifying the reviewer as one who belongs to category (i) and ignorant of category (ii). After all who made these categories? Work on asymptotic analysis and stability of stochastic approximation dates back to 1951 when the first paper on stochastic approximation came out. On the other hand, category (ii) that the authors refer to is fairly recent. And there is no reason why ideas from category (i) should not be used in category (ii). I only pointed to the Saxena et al paper for that reason because it shows not just finite time bounds on in average reward actor-critic setting  but also stability and asymptotic convergence of the algorithm. As a reviewer I have every right to ask questions to the authors and they shouldn't take these questions personally on the reviewer. Moreover, if I say that I once again went through the paper and have a couple of additional questions, again I have every right to do so as a reviewer. Why is that "disturbing" to the authors. This is the seventh paper I am reviewing for NeurIPS this year and I submitted all my reviews more than 40 days ago. So, quite obviously I don't remember all the details of every single paper that I reviewed. It does not mean that I am reading your paper for the first time.
> > > > > > >
> > > > > > > Finally, as with other reviewers, I find lack of experiments to be a clear drawback in this work.

---

> > > > > > > > ### Author Response · Authors · 2023-08-17
> > > > > > > >
> > > > > > > > We would like to know why Assumption 3(a) is strong. As we have stated, we can assure this with an arbitrarily small loss in performance. Regarding the rest of the comment, we would like to know whether the reviewer feels that showing $J^*-E(J_{\mu_k})$ is small is a good result or not. If it is not, we would like to know why since ours is the first result of this type. Stability in the sense of asymptotic boundedness a.s. is not required here because it is not a stochastic approximation proof. We have a deep knowledge of stochastic approximation theory, have worked extensively on it, and are very familiar with the beautiful theory behind the Borkar-Meyn boundedness results of that type. We have deep appreciation and respect for both (i) and (ii) style of results. To insinuate that a result of type (ii) is not interesting because the reviewer insists on results of type (i) seems very unfair. Yes, the reviewer can certainly ask questions but to not acknowledge the originality of our result or our contributions, and insisting on another type of result is what we are surprised by. As in Bhandari-Russo-Singal and Srikant-Ying, one can show the error in function approximation is bounded (either by using projection as in Bhandari et al or without using projection as in Srikant-Ying). The only assumption we need is that $\theta^*$ is bounded for all policies. This assumption essentially states that one can get bounded function approximation error with a bounded set of parameters. To the best of our knowledge, this paper presents the first convergence results under such an assumption which has been made in Agrawal et al and related papers. If the reviewer thinks that that there is an error in our proof, we would be very happy to know what it is and will be happy to address it. But we cannot answer a question if the reviewer says that they think boundedness of iterates is usually needed in the style of proofs that they are used to, without pointing out an error in our proof. Regarding simulations, the type of algorithms we have studied have been extensively studied and simulated in prior literature. Theory was what was lacking, and we believe that NeurIPS has always had a place for purely theoretical papers.
> > > > > > > >
> > > > > > > > The reason we commented on what we thought the reviewer may have had in mind while doing their review was because there was no acknowledgement of the point-by-point responses that we provided to the first review. If our understanding of the reviewer's expertise was incorrect, we apologize and are happy to stand corrected. Moving forward, we believe that we can have a productive discussion if the reviewer acknowledges the point-by-point responses to their 16 comments (11+5) and let us know which of their responses they agree with and which they do not. We will do our best to provide any required clarifications.

---

> > > > > > > > > ### Comment · Reviewer_M1AZ · 2023-08-17
> > > > > > > > > **response from M1AZ**
> > > > > > > > >
> > > > > > > > > Even though I may not have said this, but I agree that the authors have cleared many of the questions I had regarding their work. I don't see now a technical flaw in the work of the authors.  My only point is whether the results could have been stronger. Arguing along these lines, I would like the authors to refer also to the paper ``S.Chandak, V.Borkar and P.Dodhia, Concentration of Contractive Stochastic Approximation and Reinforcement Learning, Stochastic Systems, 2023,
> > > > > > > > > https://pubsonline.informs.org/doi/pdf/10.1287/stsy.2022.0097".
> > > > > > > > >
> > > > > > > > > The authors of the above paper provide finite time bounds but they are also able to show convergence with probability one under Markov noise (i.e., the asynchronous setting) for many RL algorithms and they have uniform high probability bounds with minimal assumptions. They don't use ergodicity as well (at least not aperiodicity of the process). I believe that such a result is  stronger than the results provided by the authors which is on providing finite-time bounds on the expected error. I would like to know what the authors feel about this.

---

> > > > > > > > > > ### Author Response · Authors · 2023-08-17
> > > > > > > > > >
> > > > > > > > > > This is a good question. Indeed we have thought about this. Our current paper has two contributions: (a) a general framework to analyze policy-based methods and (b) analysis of specific policy-based methods using the general framework. We don't think it would be possible to use the Chandak et al approach for (a). However, it may be possible to use it for (b), i.e., analysis of specific algorithms. A significant hurdle in this regard is the fact that the iterations cannot be written as a contractive stochastic approximation algorithm. One of the main contributions of our paper is to prove bounds despite the fact that there isn't an explicit contraction property.  Whether the Chandak et al results can be extended to more general stochastic recursions, without an explicit contraction property, is an interesting open question, which we agree is worth studying.

---

### Official Review · Reviewer_qfjY · 2023-07-04

**Soundness:** 3 good
**Presentation:** 2 fair
**Contribution:** 2 fair
**Rating:** 3
**Confidence:** 4

**Summary:**

This paper proposed an Approximate Policy Iteration Algorithm for average-reward MDPs, and further extended to RL setting.

**Strengths:**

The paper is well-written and clear. The approximated PI approach is new and novel; The theoretical results are also comprehensive.

**Weaknesses:**

The algorithm seems unclear to me. How do the updates implement (e.g., Lines 3,4 in Alg 2)?

How does the approach handle the "contraction" in the average-reward setting?

Overall, this paper seems a little bit unsurprising to me, the author should highlight the novelty and contribution in the work.

**Questions:**

See above.

**Limitations:**

It is unclear if the algorithm can be implemented in practice.

---

> ### Author Rebuttal · Authors · 2023-08-07
>
> We thank the reviewer for their review and comments. Please find our response to the weaknesses below:
>
> 1. The approximate policy iteration algorithms in the paper is a framework to study RL algorithms and is not an RL algorithm by itself. One has to then relate them to RL algorithms so that we can use bounds on policy-based RL methods by showing that they are instantiations of approximate PI. Hence the updates in lines 3,4 in Algorithm 2 are a consequence of the particular policy learning and policy improving algorithm employed. The implementation of approximate PI using RL algorithms is discussed in Section 4.
>
> 2. Average reward problems generally leverage a span contraction property due to the absence of the discount factor (which acts the source of contraction for discounted reward MDPs). However, the span contraction is not helpful in the setting of the paper, so most of the analysis involved is in circumventing this issue. Instead of contraction, we mainly leverage Lemma 2.1 and Lemma 2.2 in the appendix to deal with the lack of span contraction. Specifically, there is no contraction factor in our problem and the analysis is novel because of this fact.
>
> 3. As pointed out by the reviewer, dealing with the lack of span contraction (and the lack contraction due to discount factor) does require novelty and the specific contributions of the paper are itemized on page 2 and 3. The surprising part of the result is evident from the reviewer's previous comment which notes the lack of a contraction factor.

---

### Official Review · Reviewer_KNuh · 2023-07-04

**Soundness:** 3 good
**Presentation:** 3 good
**Contribution:** 2 fair
**Rating:** 4
**Confidence:** 3

**Summary:**

This paper focuses on finite-sample convergence analysis for average-reward reinforcement learning. The authors firstly propose finite-time  error bounds for approximate policy iteration. The major technique ingredient is performing Schweitzer transformation to the MDP. By this transformation, the stationary distribution of any policy could cover the state-space uniformly with a condition number $\gamma>0$. Therefore some contraction arguments in the analysis for discounted MDP could be applied.

**Strengths:**

1. The authors firstly achieve a finite-time error bounds for approximation policy iteration in average-reward RL with using Schweitzer transformation;

2. The authors also present a more refined analysis with respect to time-varying policy evaluation and improvement errors. Some extensions on practical algorithms (e.g, greedy and softmax) are established.

**Weaknesses:**

My major concern is about the technical novelty. The authors main contribution is to generalize the finite-time error analysis for $\alpha$-discounted MDP to the average-reward MDP, and the major technique is using the Schweitzer transformation, which I think is roughly equivalent to  learning $\lambda_1 \mathbf{I}+\lambda_2 \bar{P}+(1-\lambda_1-\lambda_2)P$ instead of learning $P$ for some proper $\lambda_1,\lambda_2\in (0,1)$, where $\bar{P}$ is the averaged transition model over all actions, and $P$ is the true transition model. In this way, the contraction arguments could work directly.

I have learned similar arguments before in literature on online regret minimization for average-reward MDP (e.g. [Wei, 2020], [Wang, 2022]), where $\lambda_1 \mathbf{I}+(1-\lambda_1)P$ is considered instead of the original transition model $P$. Moreover, [Wang, 2022] described the relationship between near-optimal policies for $\alpha$-discounted MDP and that for average-reward MDP. More precisely, it was proven that an $\epsilon$-optimal policy for $\alpha$-discounted MDP is at least $\epsilon+(1-\alpha)\mathrm{sp}(h^*)$-optimal for the average-reward MDP as long as the MDP is weak-communicating. Here $\mathrm{sp}(h^*) = 2\min_{x\in \mathbb{R}}\|h^* - r\cdot \textbf{1}\|_{\infty}$.

One can also show that $\|V^*_{\alpha}-\min_{s}V^*_{\alpha}(s) \textbf{1} - h^*+\min_{s}h^*(s)\cdot \textbf{1}\|\leq O((1-\alpha)D\mathrm{sp}(h^*))$ , where $V^*_{\alpha}$ is the optimal value function for the $\alpha$-discounted MDP and $D$ is the diameter of the MDP.
In this work, the authors introduce another term $\bar{P}$ into the transition model to make the MDP irreducible for any policy. But I do not think it is a good choice since the mixing time could be exponentially large, which might make $1/\lambda$ is exponentially large.


References:

[Wei, 2020] Model-free Reinforcement Learning in Infinite-horizon Average-reward Markov Decision Processes

[Wang, 2022] Near Sample-Optimal Reduction-based Policy Learning for Average Reward MDP

**Questions:**

1. In line 18 in the appendix, why $\hat{V}_{\mu}$ is bounded? And what is the exact bound? In general, $\hat{V}$ could be large if the mixing time of the MDP is large. Do you need $\mathrm{sp}(h^*)$ to bound $\hat{V}_{\mu}$?



**Limitations:**

Yes

---

> ### Author Rebuttal · Authors · 2023-08-07
>
> We thank the reviewer for their review, comments and references. We will add them to the paper and discuss why they do not apply to our problem. In response to the reviewer's comments, we provide the following remarks:
>
> 1. The Schweitzer transformation is not enough to show contraction. We are just using it to ensure aperiodicity. However, this alone does not provide a source of contraction in the proof of average-reward MDPs. In fact, there is no source of contraction as far we know, and our proof only shows that the minimum of the value function contracts. The maximum does not, which is why the proof is complicated. In particular, the span does not contract.
>
> 2. The Wei et al and Wang et al papers are for tabular settings. Moreover, Wei et al is for value-based algorithms (i.e., Q-learning) while Wang et al assumes access to a generative model. Specifically, Wang et al assumes $\mathbb{P}(s'|s,a)$ can be estimated by repeatedly setting the MDP to state $(s,a)$ to observe the next transition. This allows them to estimate the probability transition matrix and solve a planning problem.
>
> 3. We agree with the reviewer about the relationship between $V_\alpha^*$ and $h^*.$ But we are unable to figure out how this helps in getting error bounds for average-reward policy-based RL. Regarding the comment on the mixing time, we are not sure what the reviewer means by saying that the mixing time will be exponentially large; we are not sure exponentially in what quantity.
>
> Response to Question:
>
> $\\mathbb{P}\_{\\rho} \\widehat{V}\_{\\mu} - \\widehat{V}\_\\mu$ is upper bounded by $\\text{span}\\{\\widehat{V}\_\\mu\\}$,  where $\\widehat{V}\_{\\mu}$ is the value function associated with the modified probability transition kernel $\\widehat{\\mathbb{P}}\_{\\mu}$. Since we consider finite state-action spaces, bounded rewards and our modification ensures that all policies induce ergodic Markov chains, the $\\text{span}\\{\\widehat{V}\_{\\mu}\\}$ is well defined and bounded. Therefore, we do not need $V\_{\\mu^*}$ (referred to as $h^*$ in the question) to bound $\\widehat{V}\_{\\mu}$.

---

### Official Review · Reviewer_h7bf · 2023-07-06

**Soundness:** 4 excellent
**Presentation:** 4 excellent
**Contribution:** 4 excellent
**Rating:** 7
**Confidence:** 1

**Summary:**

This paper obtains performance bounds for policy iteration in the average reward setting, where instead of the usual discontinued factor formulation, the limiting ratio between total reward divided by time is used as a performance metric, which is a commonly used metric in the regret-based literature.The paper explains why obtaining such bounds has been difficult, and utilizes novel techniques and shows how we can obtain the bound.

**Strengths:**

- The technical claims seem sound, and the need for this bound in the average setting is well motivated.
- The authors do a good job of explaining why it has been difficult to use commonly used techniques for analyzing performance bounds for PI.

**Weaknesses:**

- It would be nice if the authors could mention meaningful problems that minimize for the average setting instead of the discounted setting. In other words, the author's mathematical contribution is well conveyed, but it would be good to motivate why we should care about the average setting in the first place despite the difficulty of analyzing them.


**Questions:**

I am not familiar with RL theory and do not have the necessary background to ask questions on the topic.

**Limitations:**

The authors have adequately addressed the limitations.

---

> ### Author Rebuttal · Authors · 2023-08-07
>
> We thank the reviewer for their review and their comments. Average reward RL formulations capture settings where long term rewards are considered more valuable than short term. Some of these applications include queuing network control, scheduling jobs during cloud computing, robot planning and control and communications networks. We are happy to include these applications in the paper.

---

> > ### Comment · Reviewer_h7bf · 2023-08-15
> > **Acknolwedgment**
> >
> > I would like to thank the reviewer for the response - unfortunately I really don't have a good background on this topic and would like to keep my current score but along with the confidence rating.

---

### Official Review · Reviewer_z18A · 2023-07-08

**Soundness:** 3 good
**Presentation:** 4 excellent
**Contribution:** 3 good
**Rating:** 6
**Confidence:** 4

**Summary:**

This paper studies the convergence of approximate policy iteration for the average reward setting. First, finite suboptimality bounds are provided for average reward approximate policy iteration under suitable ergodicity properties of the set of policy-induced Markov chains. Next, these results are extended to Q-iteration-type algorithms, building a bridge to reinforcement learning methods. Finally, using this bridge, corresponding bounds are provided for a variety of existing policy iteration-based reinforcement learning methods, including greedy policy updates, softmax policy updates, and mirror descent updates.

**Strengths:**

This paper is well-written, it builds the necessary technical background thoroughly but concisely, its contributions and results are clearly stated, and it addresses a longstanding open issue in classical policy iteration methods for average-reward MDPs. The overall plan of the paper is to rigorously establish a connection between average-reward approximate policy iteration methods and policy iteration-based average-reward reinforcement learning methods, inspired by the work [Z. Chen & S. T. Maguluri, _Sample complexity of policy-based methods under off-policy sampling and linear function approximation_. AISTATS, 2022], which achieved this for the discounted setting. In the present work, this is achieved largely via an analysis based on [J. Van der Wal,  _Successive approximations for average reward Markov games_. International Journal of Game Theory, 1980], but suitably modified to handle the presence of approximation errors in the policy recovery and policy evaluation steps. I did not check the appendix in detail, but the general thrust of the arguments appears to makes sense. The work has a somewhat "classical" flavor, as access to the transition dynamics, rewards, and approximate policy improvement and evaluation oracles appears to be assumed in the algorithms and analysis, but it should nonetheless be of interest to the theoretical RL community, especially those interested in RL policy iteration and policy evaluation underpinnings.

**Weaknesses:**

My main concerns are the following:
* The apparent access to transition dynamics, rewards, and approximate policy improvement and evaluation oracles will likely appear unreasonable to those unfamiliar with, e.g., [D. Bertsekas & J. N. Tsitsiklis, _Neuro-dynamic programming._ 1996]. For this reason, a discussion of how the approximate policy improvement and approximate policy evaluation steps can be carried out in practice would help allay this concerns and strengthen the paper.
* The key innovation and insight in the analysis remains somewhat unclear. Lines 119-125 in the appendix mention that the analysis of [Van der Wal, 1980] must be modified by carefully bounding the errors arising from approximate policy evaluation and improvement, but it is still unclear what the magnitude of the effort was and whether any specific, innovative techniques were used.

**Questions:**

* How can the approximate policy evaluation and improvement oracles be instantiated in practice?
* Did the careful bounding mentioned in lines 119-125 of the appendix consist solely of manipulations of inequalities and algebraic expressions? Or was there some key, interesting insight or innovative application of some unlooked-for technique required?

**Limitations:**

Aside from the weaknesses and questions above, the authors have addressed the limitations.

---

> ### Author Rebuttal · Authors · 2023-08-07
>
> We thank the reviewer for their review and their comments. Please find our response to the weaknesses/questions below:
>
> 1. Section 4 deals with how RL algorithms implement approximate PI. Approximate PI by itself should not viewed as an algorithm. The main novelty of the paper is show how one can analyze many practically implementable RL algorithms as instantiations of approximate PI. Therefore, analyzing approximate PI provides a path to analyzing policy-based algorithms, which is the main message of the paper. Specifically, we do not require access to transition dynamics, rewards, and approximate policy improvement and evaluation oracles because we are not directly implementing approximate PI. We are only showing that many RL algorithms naturally fit into this framework and these algorithms do not require access to approximate policy evaluating or approximate policy improving oracles.
>
> 2. The idea behind relating consecutive iterates of the algorithms was inspired by Van der Wal [1980], however the rest of the analysis is mathematically different. Since Van der Wal studies modified policy iteration, there is no explicit notion of error in the policy improvement or policy evaluation steps. However, such errors are introduced in RL algorithms and therefore, to study RL algorithms, we have to study approximate policy iteration rather than modified policy iteration. As a result, it is important to make sure that the errors do not propagate from one iteration to another. In fact, this is the reason that the analysis for the discounted case fails to yield a non-trivial bound for average-reward problems in the limit the discount factor goes to one. The novelty in our analysis is to carefully bound the propagation of the error. We invite the reviewer to compare the steps of analysis in pages 9-11 of Van der Wal and our analysis in the proof of Theorem 3.3 to see the differences. Additionally, we note that the only prior works that were able to get sublinear regret were [AYBB+19] and [LYAYS21]. However, they were not able to analyze practical policy evaluation techniques such as TD-learning and other policy improvement algorithms (such as greedy update and softmax update) because of the fact that they were exploiting a connection to learning from experts. We believe that introducing the technique of Van der Wal [1980] to study RL algorithms (which to the best of our knowledge has never been done before) is a significant novelty in itself. Recognizing that the technique applies more broadly to RL algorithms (and not just to planning problems which was Van der Wal's motivation) is innovative application of a previously unknown technique to study RL algorithms.

---

> > ### Comment · Reviewer_z18A · 2023-08-18
> >
> > Thanks for your response. My concerns regarding how the approximate policy improvement and approximate policy evaluation steps can be carried out in practice have been allayed by response 1. Response 2 has partially clarified the key innovation and insight in the analysis in your paper by emphasizing the importance of your extension of the technique used by [VdW80] to handle propagation of errors over time. However, it is still unclear (a) how the analysis of [VdW80], which is for a two-player game, is modified for the MDP setting of this paper, and (b) how challenging incorporating errors is in this extension. Also, since introducing the [VdW80]-inspired technique into RL appears to be a core contribution, it would be helpful to see a more thorough discussion of all of the foregoing in the main body of the paper. I still think this paper is a solid contribution and is of interest to the theoretical RL community and will thus keep my current score, but increase my confidence.

---

> > > ### Author Response · Authors · 2023-08-18
> > >
> > > Thanks for your comment. The key idea that we have used from [VdW80] is a method to handle the fact there is a lack of contraction in our problem: even the span of the value function does not appear to contract as in proofs of some average-reward problems. So what we have borrowed from [VdW80] is the high-level idea of showing that the minimum element of the value function vector increases monotonically and further, by showing that even though the maximum element does not decrease monotonically, we can nevertheless show that asymptotically the minimum and maximum elements converge towards each other. Indeed the details are different than in [VdW80] and we will add a discussion in the main body of the paper. It is a bit challenging to explain how difficult it was to incorporate errors in the analysis since almost all the steps of [VdW80] have to be changed (other than the high-level idea) to establish our proof. However, we will continue to think about how to concisely explain this in the paper.

---

### Official Review · Reviewer_WTrY · 2023-07-13

**Soundness:** 3 good
**Presentation:** 4 excellent
**Contribution:** 3 good
**Rating:** 6
**Confidence:** 4

**Summary:**

The paper "Performance Bounds for Policy-Based Average Reward Reinforcement Learning Algorithms" tries to solve the open problem of obtaining meaningful performance bounds for approximate policy iteration algorithms for the average reward setting. If one tries to use the bounds for PI from discounted reward setting and set the discount factor equal to one to obtain the average reward setting, due to the horizon dependence, the bounds tend to infinity. While this is so, it is well known that API performs better in practice than this bound suggests. This paper tries to bridge this gap between theory and practice. It first obtains finite time error bounds for average reward approximate policy iteration. Then, it extends to obtain finite-iteration error bounds for the case where the errors can vary over iterations. Later, it illustrates the use of these results for various policy-based RL methods.

**Strengths:**

1. Directly using the bounds for PI from discounted reward setting by setting the discount factor equal to 1 and obtaining the average reward setting, the bounds tend to infinity. This is due to the horizon dependence.  From the literature, it is well known that API performs well in practice, in contrast to what this bound suggests. This paper tries to bridge this gap between theory and practice by presenting meaningful performance bounds for approximate policy iteration algorithms for the average reward setting.

2. Their bounds show that the error goes to zero, unlike the transformation of bounds from discounted to average reward.

3. The motivation is clear and the paper is very well written.

4. Obtained performance bounds for various RL based methods by plugging the above bounds.

**Weaknesses:**

1. In the performance bounds for RL based methods, the errors don't seem to be small. They all depend on $\gamma$, which can make them worse. (This is discussed in remark 4.6, but I don't understand the use of these results if this is the case, please clarify this).

2. In section 4.2, the regret of an online algorithm is discussed. In equation $16$, the regret is of the linear order, while the earlier work has $\sqrt{K}$ dependence. Moreover, this bound depends upon $\gamma$. Then again, what is the use of these performance bounds?

A minor issue:

$\gamma$ is not clear or defined before Theorem 3.3.

**Questions:**

Please provide clarifications to the questions above.

**Limitations:**

Yes, the authors have addressed the limitations.

---

> ### Author Rebuttal · Authors · 2023-08-07
>
> We thank the reviewer for their review and their comments. Please find our response to the weaknesses below:
>
> 1. Our results state that in the limit when the function approximation error goes to zero, the learned policy's performance approaches the performance of the optimal policy. Such a result was previously non existent in prior literature on average reward MDPs, except in a very special case considered in [AYBB+19, LYAYS21]. We will comment on [AYBB+19] in this bullet and will comment on [LYAYS21] in the next bullet.  In the context of mirror descent with modified LSPE [AYBB+19] provide with $O(K^{3/4})$ regret bounds which are independent of $\gamma$ since they leverage properties specific to mirror descent. If we were to use mirror-descent-specific properties like in their paper, we can get the same $\gamma$ independent result with the same regret bound as in paper even for TD learning. However, our result improves on the regret bound by obtaining $O(K^{2/3})$ regret but a dependence on $\gamma$. Which bound is better depends on the relative sizes of $K$ and $\gamma.$ Beyond mirror descent, it appears unlikely that we can remove the dependence of $\gamma$ since it appears even in the analysis of modified policy iteration for average-reward MDPs when the probability transition matrix and rewards are known.
>
> 2. Both the regret in our paper and [LYAYS21] have a linear component due to function approximation error, see Equation 7 in [AYBB+19] and Assumption 6.1 in [LYAYS21]. The $\sqrt{K}$ result in [LYAYS21] is the $\\textbf{additional}$ error beyond the function approximation. Our corresponding bound is $O(K^{2/3}).$ However, it should be noted that the result in [LYAYS21] is for Monte Carlo policy evaluation which is infeasible in practice (for example, due to the need to collect and store large trajectories and the need for a very large matrix inversion) while our result is for TD-learning, which is widely used. The correct comparison of our result would be to the result in [AYBB+19]; in comparison to this paper, we have improved the regret from $O(K^{3/4})$ to $O(K^{2/3}).$
>
> Minor Issue: $\gamma$ has been defined in Lemma 3.2 prior to Theorem 3.3. We will add a reference to the definition of $\gamma$ in Theorem 3.3 as well.

---

> > ### Comment · Reviewer_WTrY · 2023-08-16
> >
> > I thank the authors for answering my questions, but in view of the other considerations for the experiments, keeping the score unchanged.

---

### Decision · Program_Chairs · 2023-09-21

**Decision:**

Accept (poster)

**Comment:**

This paper contributes important advancements to the convergence theory of policy iteration in average-reward MDPs by introducing novel game-theoretic techniques. Moreover, the authors have substantively addressed all concerns by the reviewers who voted to reject the paper with high confidence, except for one who failed to reply to the discussion period completely. I recommend this paper be accepted.